

# Recent observations and glacier modeling point towards near complete glacier loss in western Austria (Ötztal and Stubai mountain range) if 1.5°C is not met

Lea Hartl[1, 2, *], Patrick Schmitt[3, *], Lilian Schuster[3], Kay Helfricht[1, 4], Jakob Abermann[5], and Fabien Maussion[3, 6]

[1]Institute for Interdisciplinary Mountain Research, Austrian Academy of Sciences, Austria
[2]Alaska Climate Research Center, Geophysical Institute, University of Alaska Fairbanks, USA
[3]Department of Atmospheric and Cryospheric Sciences, University of Innsbruck, Austria
[4]Hydrological Service Tyrol, Office of the Tyrolean Government, Austria
[5]Department of Geography and Regional Science, Graz University, Austria
[6]School of Geographical Sciences, University of Bristol, Bristol, UK
[*]These authors contributed equally to this work.

**Correspondence:** Lea Hartl (lea.hartl@oeaw.ac.at), Patrick Schmitt (patrick.schmitt@uibk.ac.at)
These authors contributed equally to the work

**Abstract.** Most glaciers in Austria are expected to disappear in the coming decades. The general trend to deglaciation is apparent from observations of past glacier change as well as projections of future glacier evolution in the region. However, the projected timing of ice loss varies considerably between models and data sources. We enhance observations of regional glacier area and volume change with a new inventory for the Ötztal and Stubai range in western Austria and use this data to

initialize and calibrate the Open Global Glacier Model (OGGM), generating projections for all glaciers in the study region until 2100 under different climate scenarios. Observations show that approximately 19% of glacier area and 23% of glacier volume were lost between 2006 and 2017 (values are relative to 2006 area and volume and equivalent to annual loss rates of 1.7% and 2.1%, respectively). Five glaciers disappeared between 2006 and 2017 and are no longer included in the 2017 inventory. Estimating future change by extrapolating the change rates observed between 2006 and 2017 produces a considerably slower

glacier decline than the model projections for all scenarios, highlighting the need for dynamic, climate-aware glacier models to quantify the range of possible futures and trajectories to deglaciation. By adapting OGGM to incorporate the multitemporal, high-resolution observational data available for the study region, the model performance improved compared to using global, lower resolution data and, for the first time, enabled the model to simultaneously match observed area and volume changes at a regional scale. This increases confidence in the regional projections, which show 2.7% of the 2017 glacier volume in the region

remaining by 2100 in a global warming scenario of +1.5°C above pre-industrial temperatures. Applying a +2°C scenario, this volume is reached around 30 years earlier and deglaciation is near complete by 2100 (0.4% of 2017 volume remaining). Gepatschferner, the largest glacier in the region, is expected to retain 5.4% of its 2017 volume in a +1.5°C scenario and 0.4% in a +2°C scenario. Over 100 glaciers, i.e. roughly one third of the glaciers in the study region, are likely to disappear by 2030 even in the +1.5°C scenario. Glacier loss in the study region under current warming trajectories (+2.7°C) is expected to be near

total before 2075 (less than 1% of 2017 volume remaining).



# 1 Introduction

Mountain glaciers are receding rapidly at a global scale (Zemp et al., 2015; Hugonnet et al., 2021a), with wide-ranging impacts on water resources (e.g. Akhtar et al., 2008; Baraer et al., 2012; Huss, 2011; Huss and Hock, 2018), societal and ecological

systems (e.g. Cannone et al., 2008; Carey et al., 2017; Bosson et al., 2023), and sea level rise (Radić and Hock, 2011; Hock et al., 2019; Zemp et al., 2019; Rounce et al., 2023). Regionally, trends in glacier area and volume vary depending on temperature and precipitation variability (Hugonnet et al., 2021a) as well as local factors (e.g. Brun et al., 2017). Remote sensing studies show that glaciers in the European Alps have lost on average between 0.70±0.13 and 1.02±0.21 m w.e. per year in recent decades (values refer to the periods 2000-2014 (Sommer et al., 2020) and 2000-2019, (Hugonnet et al., 2021a), respec-

tively). Assessments of glacier change at smaller scales find loss rates generally in the same order of magnitude though with considerable temporal and spatial variability. Swiss glaciers lost on average between -0.52 m and -1.07 m of elevation per year between 1980 and 2010 depending on the catchment (Fischer et al., 2015c). Relative area change of Swiss glaciers also varies considerably with catchment, topography, and glacier size class, with largest losses at smaller glaciers (Linsbauer et al., 2021). Glacier area decreased by 1.2% per year between 1998 and 2006 in Austria (Fischer et al., 2015b). Similar area change rates

have been reported for France (Gardent et al., 2014) and the entirety of the European Alps (Paul et al., 2019).

Observations of glacier change are used to drive models that project the future evolution of glaciers. The compilation of the Randolph Glacier Inventory (Pfeffer et al., 2014; Randolph Glacier Inventory Consortium, 2017, RGI,) enabled modeling at large scales by reducing the uncertainties associated with extrapolation or upscaling of model data to un-inventoried glaciers (Hock et al., 2019). Despite considerable advances in large-scale modeling (Marzeion et al., 2012; Huss and Hock, 2015;

Zekollari et al., 2019; Maussion et al., 2019; Cook et al., 2023), the Glacier Model Intercomparison exercises (GlacierMIP 1 and 2) found significant differences between glacier model projections particularly at the regional scale, which stem from differences in models physics, calibration and setup, and from differences in input data (Hock et al., 2019; Marzeion et al., 2020). Uncertainty originating from modeling choices constitutes the largest relative contribution to overall uncertainty for projections until about mid-century and remains considerable afterwards (Marzeion et al., 2020).

While the GlacierMIP studies cannot disentangle between the various sources of uncertainty, the large uncertainties in projections for the next decades hint at the importance of accurate boundary conditions to calibrate and initialize glacier models. The current approach for large-scale models usually relies on a single glacier inventory (e.g., the RGI), an ice thickness estimate to initialize the model, and one mass change product to calibrate model parameters. When these are used consistently across models, differences in the first decades are considerably reduced (Zekollari et al., 2024), underscoring the importance

of observations to constrain glacier projections.

For regional modeling, it is therefore crucial to first quantify recent regional glacier change. High resolution regional inventories can delineate spatial variability of change rates within the same mountain range and improve understanding of local topographic and climatic factors that drive such variability (Fischer et al., 2015c; Linsbauer et al., 2021). In regions with many





small glaciers and rapid glacier loss, such as Austria and Switzerland, a substantial fraction of the total ice mass is contained
in small, increasingly debris-covered features, that are difficult to accurately map based solely on medium-resolution optical
satellite imagery (Fischer et al., 2014, 2021). Incorporating high resolution topographic data in updates to glacier area and
volume change data sets improves the spatio-temporal coverage and granularity of the time series. Additionally, such data con-
tributes to uncertainty reduction in larger scale, global or Alps-wide inventories, e.g. by providing better constrained baseline
data on the location of ice divides and debris cover change (Paul et al., 2019).

We aim to contribute to an improved understanding of local and regional glacier change in western Austria. Specifically,
we quantify past glacier area and volume change for the time period 2006–2017/18 in the Ötztal and Stubai Alps based on
digital elevation models (DEM) generated from airborne laser scanning survey data, extending the existing time series of
regional glacier inventories. We then use these high resolution regional datasets to dynamically calibrate and validate a glacier
evolution model over several decades, a novel approach in large-scale modeling. We project glacier changes until 2100 for
different global temperature scenarios. By using regional data (Section 2.1.1) to initialize and calibrate model projections, we
aim to better constrain possible timelines to deglaciation in the study region. We contrast our findings with projections driven
with global data sets (Section 2.1.2) to assess the impact of different calibration data and starting conditions on model outcome.

## 2 Methods and Data

### 2.1 Study region and previous studies

The Stubai and Ötztal mountain ranges are located in western Austria and border on Italy along the main Alpine divide (Fig.
1). Glacier coverage extends from around 2400 m.a.s.l. to the regions' highest peak (Wildspitze) at 3768 m.a.s.l. Climatolog-
ically, the Stubai and Ötztal Alps are characterized by dry inner-alpine valleys with strong elevation gradients of temperature
and precipitation. Annual precipitation increases from around 800 mm $yr^{-1}$ in the relatively dry valleys to more than 1500
mm around 3000 m a.s.l. (Strasser et al., 2018). Glaciological research in the area has a long tradition (e.g. Finsterwalder,
1897; Hoinkes and Rudolph, 1962; Ambach and Eisner, 1966; Hoinkes, 1969; Strasser et al., 2018) and three World Glacier
Monitoring Service (WGMS) reference glaciers are located within the study region - Hintereisferner (HEF), Kesselwandferner
(KWF), and Vernagtferner (VF).

All available observational data from the study region indicate volume loss and a rapid decrease of glacier area (e.g. Aber-
mann et al., 2009; WGMS, 2023). In large-scale projections of future glacier evolution, a complete or near-complete loss of
present day ice in the study region - and the European Alps in general - is anticipated under current climate trajectories (Hanzer
et al., 2018; Zekollari et al., 2019; Compagno et al., 2021; Rounce et al., 2023). However, the timing of deglaciation and initial
ice volume estimates vary considerably between studies (see discussion).



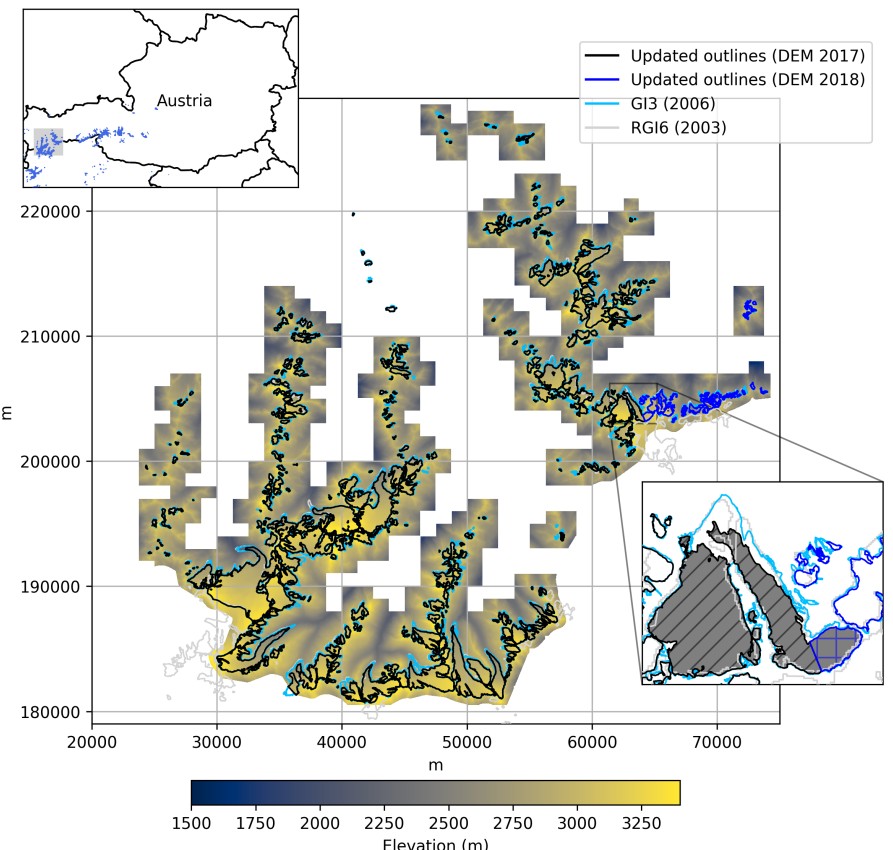

**Figure 1.** Upper inset: Location of the study area in Austria (grey rectangle, with RGI6 glacier outlines in blue). Main panel: The 2017/18 glacier outlines plotted over the 2017/18 DEM mosaic used in this study. For black outlines, the underlying DEM was acquired in summer of 2017. For blue outlines, the DEM was acquired in summer of 2018. The lower inset shows the boundary between the 2017 and 2018 DEM where it crosses Sulzenauferner (grey shading shows 2017/18 outline of Sulzenauferner). The light grey lines indicate the RGI6 glacier outlines. Coordinate reference system: MGI / Austria GK West, epsg: 31254.

### 2.1.1 Regional data sets: Area, volume, volume change

Multiple regional glacier inventories (GI) quantify the evolution of glacier extent since the Little Ice Age (LIA). Regional
volume change has been assessed in irregular intervals based on digital elevation models (DEMs), which were generated
during the compilation of the first and second Austrian Glacier Inventories (GI1 (Patzelt, 1980), GI2 (Lambrecht and Kuhn,
2007)) or made available by the regional government of Tyrol in more recent years (GI3 (Fischer et al., 2015a) and this study).
GI1, GI2, GI3, as well as the LIA GI (Groß and Patzelt, 2015) were compiled with the aim of consistency between inventories
in terms of the mapping approach and glacier identification. Outlines were mapped based on the DEMs, taking elevation change
information into account. An additional inventory representing glacier extent in 2015 (GI4, Buckel and Otto (2018)) derived





from Google Earth imagery shows minor inconsistencies with previous GI due to differences in the underlying data. Table 1 and Table 2 give an overview of relevant datasets with citations of previous work as well as data publications where applicable.

GI3 counts 322 glaciers covering 187 km$^2$ in 2006 in the region of interest (ROI, Fig. 1). A high resolution estimate of ice volume in the region is available for the same year from Helfricht et al. (2019a) (see Section 2.3) and indicates a total ice volume of 8.5 km$^3$. The glaciers in the ROI represent about 45% of the total glacierized area (Fischer2015, GI3) and 53% of the total ice volume (Helfricht et al., 2019a) in Austria (values refer to 2006).

### 2.1.2 Global data sets

We additionally extracted area, volume, and volume change data for the ROI from global data sets, namely glacier area from the RGI6 (Randolph Glacier Inventory Consortium, 2017; Pfeffer et al., 2014; Randolph Glacier Inventory Consortium, 2023), glacier mass change from Hugonnet et al. (2021a), and glacier volume from Farinotti et al. (2019); Millan et al. (2022) and Cook et al. (2023). Different methods and underlying data result in differences between the regional and global data sets. The number of glaciers and the glacierized area as per the various inventories are summarized in Table 1. Compared to the regional estimate of ice volume in the ROI by Helfricht et al. (2019a), the estimates in the larger region studies of Millan et al. (2022), Farinotti et al. (2019) and Cook et al. (2023) are higher with 9.3 km$^3$, 10.3 km$^3$ and 8.9 km$^3$, respectively (Table 3).

A common challenge for all currently available large-scale ice thickness products is to associate a timestamp to the computed volumes because of the mismatch between the various datasets used as boundary conditions and for calibration. Farinotti et al. (2019) used a DEM from 2003 in the Alps (SRTM) as well as the RGI6 outlines from 2003, but the thickness observations used to calibrate the models and compute uncertainties have a wide range of timestamps. The Farinotti et al. (2019) product is often associated with the timestamp of the RGI6 outlines. Helfricht et al. (2019a) use better coinciding data products and associate a volume to the year 2006. Millan et al. (2022) and Cook et al. (2023) use RGI6 outlines, ice velocity products dated to 2017, and different DEMs. The direct comparison to Farinotti et al. (2019) in Millan et al. (2022) indicates that the chosen timestamp of the product is 2003. However, according to the supporting information of Cook et al. (2023), Millan et al. (2022) should be dated to 2017 (the year of the ice velocity observations). Cook et al. (2023), in turn, use the velocity observation of Millan et al. (2022) for their volume estimate, but use it to initialize simulations at the year 2020 (see supp. Fig. S11).

### 2.2 Updated regional glacier outlines and DEM time series

The most recent digital elevation model (DEM) covering the study area was generated from airborne laser scanning (ALS) survey data acquired in 2017 and 2018 and processed by the department for geoinformation of the Austrian state of Tyrol. The mean ALS point density in mountainous areas is between 5.7 and 6.9 points per m$^2$ (Rieger, 2019). Comprehensive information on general processing steps, DEM generation, mosaicing, and deviations from control areas can be found in Rieger (2019). To assess glacier change, we used a mosaiced version of the 2017/18 DEM with a spatial resolution of 1m x 1m in the Austrian national grid, as available through the geodata portal of the state of Tyrol. The mosaiced DEM covers >99% of the glacier area in the study region. 97% of the glacier area was surveyed during several flights between July and October, 2017. The remaining area at the eastern edge of the study region was surveyed during three days in late July 2018 (Fig. 1). We note that the DEM



**Table 1.** Overview of glacier outlines previously available for the study region. Abbreviations: ALS: Airborne Laser Scanning, GI: Glacier Inventory. RGI: Randolph Glacier Inventory, DEM: Digital elevation model.

| Inventory (data set release date) | Year boundaries | Data basis | Number of glaciers in ROI | Glacier area in ROI (km$^2$) | References |
|---|---|---|---|---|---|
| GI LIA (2015) | 1850 | Moraines, historic maps | 249 | 376.6 | Fischer et al. (2015b); data set: Groß and Patzelt (2015) |
| GI1 (2013) | 1969 | Digitized from maps, orthophotos | 318 | 241.4 | Patzelt (1980); Groß (1987); data set: Patzelt (2013) |
| GI2 (2015) | 1997 | Orthophotos | 326 | 205.1 | Eder et al. (2000); Lambrecht and Kuhn (2007); Kuhn et al. (2012); data set: Kuhn et al. (2015) |
| RGI6 (2018) | 2003 | Landsat optical imagery in the ROI, other data sources elsewhere | 292 | 173.1 | Pfeffer et al. (2014); data set: Randolph Glacier Inventory Consortium (2017) |
| GI3 (2015) | 2006 | ALS DEM, orthophotos | 322 | 186.6 | Abermann et al. (2009, 2010); Fischer et al. (2015b); Land Tirol Abteilung Geoinformation (2011); data set: Fischer et al. (2015a) |
| GI4 (2018) | 2015 | Google Earth imagery | 369 | 149.6 | data set: Buckel and Otto (2018) |

does not extend into Italy, limiting our analysis to the Austrian part of the Ötztal range. The DEM for 2006, the previous time
step in the DEM time series, was also generated from ALS data and provided by the state of Tyrol in the same format, reference
system, and resolution as the 2017/18 data. We refer to previous work for citations detailing the 2006 DEM, as well as the older
DEMs (Table 2).

We updated the glacier outlines for the study region for 2017/18 based on the above DEMs and the 2006 glacier outlines
(Table 1). The mapping workflow followed the method presented by Abermann et al. (2009). Hillshades and slope and aspect
rasters extracted from the 2017/18 DEM were used as the basis for glacier delineation (Fig. 2). A surface elevation change raster
showing the difference between the 2017/18 and 2006 DEMs was displayed semi-transparently on top of the hillshade during
the mapping process. Debris covered ice can be hard to identify in optical imagery without additional information (Fischer



**Table 2.** Overview of regional DEMs available in the ROI. Abbreviations: ALS: Airborne Laser Scanning, GI: Glacier Inventory.

| Source | Year DEM | Data basis | References |
|---|---|---|---|
| GI1 | 1969 | Digitized from maps, orthophotos | Patzelt (1980); Groß (1987) |
| GI2 | 1997–2002 | Orthophotos | Eder et al. (2000); Lambrecht and Kuhn (2007); Kuhn et al. (2012) |
| GI3, State of Tyrol | 2006 | ALS | Abermann et al. (2009, 2010); Fischer et al. (2015b); Land Tirol Abteilung Geoinformation (2011); DEM available from tirol.gv.at |
| 2017/18 state-wide survey, Tyrol (this study) | 2017/18 | ALS | Rieger (2019); DEM available from tirol.gv.at |

**Table 3.** Overview of gridded ice volume and volume change estimates available for the ROI. Ice volume is computed as the sum over all pixels in the ROI bounding area (Fig. 1 multiplied by the pixel size.)

| Source | Coverage | Resolution | Year | Ice volume in ROI ($km^3$) |
|---|---|---|---|---|
| Helfricht et al. (2019a, b) | Regional | 10m x 10m | 2006 | 8.45 |
| Farinotti et al. (2019); Farinotti, Daniel (2019) | Global | 25m x 25m | 2003 | 10.30 |
| Millan et al. (2022) | Global | 50m x 50m | 2017 (see text) | 9.28 |
| Cook et al. (2023) | Alps | 100m x 100m | 2020 (see text) | 8.94 |

et al., 2014, 2021) and including surface elevation change in the mapping of glacier area aids the detection of ice bodies below debris cover and, hence, accurate mapping of the respective glacier outlines. Incorporating small scale geomorphological

features such as distinct local discontinuities in aspect and slope as well as the gradient of surface elevation change enables improved delineation of ice bodies and periglacial surfaces at the glacier margins (e.g. Abermann et al., 2009; Fischer et al., 2021). Orthophotos were used in small subregions where no ALS data exist (e.g. for Hochjochferner, which extends partially into Italy) and for a visual comparison of mapping results where appropriate.

The 2017/18 glacier outlines were generated by reshaping the GI3 (2006) glacier outlines (Tab. 1) in ArcMap GIS software.

The 2017/18 outlines were edited so that they are completely inside the GI3 boundaries. This ensures that any area changes




potentially originating from the correction of mismatches in the former inventories (GI2, GI3) are excluded from the most recent change analysis (GI3-2017/18, Fig. 2). We note that this is not appropriate in times of glacier advance.

Glaciers and detached parts of glaciers were excluded from the 2017/18 inventory if no surface elevation changes could be detected inside the former (GI3) boundaries. This can lead to the inclusion of dead ice or glacio-morphological features that
showed surface change and, hence, signs of ice ablation, during part of the 2006–2017/18 time period even if said features may have melted by the end of the period. The contribution of such "former glacial features" to the total glacier area is minimal and they will be detected and removed during the compilation of the next inventory.

Former glaciers or glacier parts which showed no clear surface elevation change related to ice melt were deleted from the inventory (Fig. 2). No discrete exclusion based on a minimum size threshold was implemented. The uncertainty in glacier
area due to mapping errors is estimated to be $\pm1.5\%$ for glaciers larger than 1 km$^2$ and $\pm5\%$ for smaller glaciers following Abermann et al. (2009) and Abermann et al. (2010). These values represent the upper bound of uncertainty estimates based on mapping of the same glaciers by multiple parties and accounting for terrain-related ambiguities. Small deviations due to errors in GI2 and GI3 are covered by the uncertainty estimation in glacier area. The 2017/18 outlines produced as part of this study are available as data publications (Helfricht et al., 2024a, b). Area change uncertainty was computed as the root-sum of the
squares of area uncertainties in the two time steps being compared.

Volume change for all glaciers in the study region was computed based on DEM differencing. The 2017/18 and 2006 DEMs were resampled to a 5mx5m resolution to match the resolution of older DEMs and generate a consistent time series of difference rasters. All DEMs in the time series were coregistered to the 2017/18 DEM following the approach of Nuth and Kääb (2011) as implemented in the python package xDEM (xdem contributors, 2021). Difference rasters were clipped with the
glacier outlines pertaining to the respective older DEM, i.e., we used the glacier outlines of 2006 to compute elevation change for the 2006–2017/18 time period. Thickness change was computed on a pixel-wise basis and integrated over the glacier area to obtain volume change for individual glaciers, elevation bins, glacier size classes, and the entire study area. Mean annual change rates for each DEM pair were computed by dividing total change by the number of years between DEM acquisitions. For the 2006–2017/18 difference raster, this results in mean change rates for 2006–2017 in areas surveyed in 2017, and mean change
rates for 2006–2018 for areas surveyed in 2018. One glacier - Sulzenauferner in the Stubai region - was surveyed partially in 2017 and partially in 2018 (Fig 1). In this case we use the 2006–2017 mean change rate for further analysis. Hochjochferner on the southern edge of the ROI extends partially into Italy and elevation data is not available for 2017 for the Italian part of the glacier. The outline was mapped using orthophotos but the volume change information is incomplete. Based on the geometry of the glacier, we exclude the westernmost, partially Italian sector of Hochjochferner from further analysis to avoid
inconsistencies due to the lacking DEM.

As a measure of relative precision of the DEM time series, we assessed stable terrain outside of glacier areas following the methods described in Hugonnet et al. (2022) to account for elevation heteroscedasticity. For each time period and respective DEM pair, slope and curvature were derived from the more recent DEM and heteroscedasticity was subsequently inferred (xdem contributors, 2021; Hugonnet et al., 2022). The estimated standard error of the mean elevation for a range of slope
angles and curvatures is given in Table 4. For the high resolution, quality controlled 2006–2017/18 ALS DEM pair, terrain



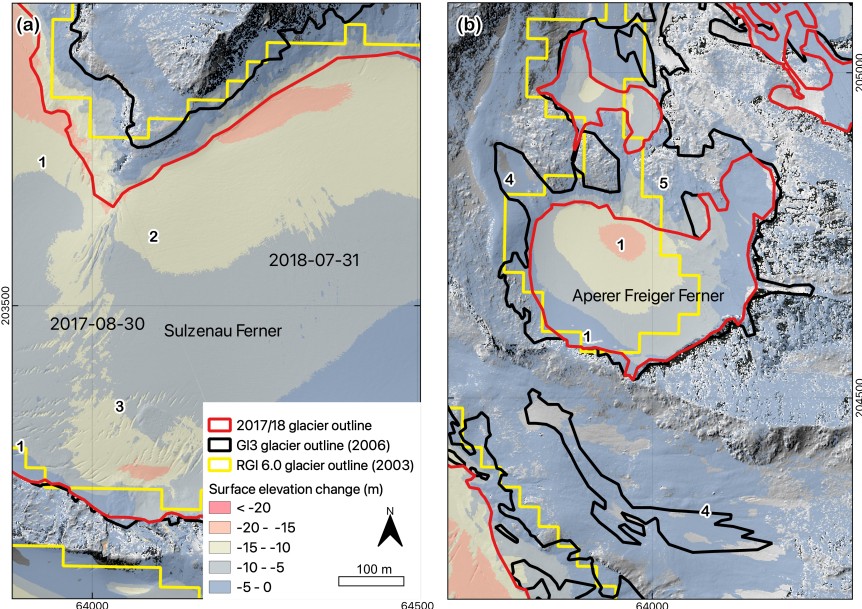

**Figure 2.** Examples of challenges related to the mapping of glacier outlines: 1) gradual change of surface elevation changes (dz) towards strong changes with maximum dz at the lower glacier margins and clear delineation towards zero dz of rock faces. 2) dz offset (visible line) caused by the different acquisition dates as labeled in the figure. 3) dz caused by processes other than ice melt, e.g rock fall deposition. 4) Former glacier parts which show no clear surface elevation change related to ice melt and were deleted in the updated inventory. 5) Errors in former inventories: in this case bare rock areas without any dz were included and ice-covered areas were excluded in GI3. This was corrected in the new inventory.

dependent error variability is low and errors do not typically exceed 0.2 m even for steeper slopes. This estimate is in line with previous work assessing uncertainties between ALS DEMs in the study area and adjacent mountain ranges (Abermann et al., 2010; Fischer et al., 2021; Hartl et al., 2022). The normalized absolute median deviation (NMAD) for stable terrain outside of the glacier boundaries was computed as an additional estimate of overall accuracy of the DEM differencing.

To assess the influence of the elevation errors on elevation change at individual glaciers, the errors inferred from stable terrain were spatially propagated based on variograms estimated from standardized elevation differences and the respective glacier outlines, again following Hugonnet et al. (2022). Uncertainties in mean elevation change due to DEM elevation errors are minimal for the 2006–2017/18 DEM pair and - like the elevation errors - show little terrain dependency. For the 1997–2006 DEM pair, uncertainties are in the range of 1.0-1.6m and increase with slope angle (supp. Fig. S1). For estimates of the

uncertainty of volume change at individual glaciers or aggregated over size classes or the ROI, we assume that the errors in glacier area and elevation change are independent (Hugonnet et al., 2021a).



**Table 4.** Mean estimated elevation error (m) derived from stable terrain for different slope angles and curvatures for the 1997–2006 and 2006–2017/18 DEM pairs.

| Slope (°) | maximum curvature (m$^{-1}$) | 1997–2006 | 2006–2017/18 |
|:---:|:---:|:---:|:---:|
| 0 | 0 | 2.12 | 0.16 |
| 30 | 0 | 2.48 | 0.16 |
| 40 | 0 | 2.92 | 0.18 |
| 0 | 0.05 | 2.12 | 0.16 |
| 30 | 0.05 | 2.48 | 0.16 |
| 40 | 0.05 | 2.92 | 0.19 |

## 2.3 Ice volume

The ice volume for each glacier in the study area was computed by adding the volume change derived from DEM differencing for the 2006–2017/18 period to the regional, 10 m resolution gridded ice thickness product for 2006 (Helfricht et al., 2019a, b). The ice thickness model underlying the distributed product was calibrated with ice thickness measurements from 58 glaciers. The mean error of the dataset is given as 25-31% based on a comparison of measured and modeled point ice thickness. In the following, the 2006 ice volume estimate serves as the starting point for calculating projected future evolution of absolute ice volume in the study area.

For a "constant change rate" future scenario based on 2006–2017/18 volume change rates, the mean annual change rate was computed on a per-pixel basis. The resulting change rate was subtracted from the total volume in annual time steps to produce extrapolated ice volume rasters until the year 2100.

## 2.4 Open Global Glacier Model (OGGM) projections

For the assessment of glacier evolution until 2100 under a range of possible future climates, we conducted dynamic glacier model simulations using the Open Global Glacier Model OGGM (Maussion et al., 2019). OGGM uses a monthly temperature index model to compute the climatic mass balance and an ice dynamical numerical model based on the shallow ice assumption to compute mass flow along 1D elevation band flowlines. In this study, we used OGGM v1.6.1, which is briefly described in Zekollari et al. (2024) and comprehensively documented in Maussion et al. (2023), https://docs.oggm.org/en/v1.6.1/.

A first set of simulations, referred to as OGGM default, corresponds to the OGGM standard simulations described in Zekollari et al. (2024) and shared publicly in Schuster et al. (2023b). Like other large-scale glacier models, OGGM default relies on global datasets of glacier area and volume as the basis for future projections (e.g. Zekollari et al., 2019; Compagno et al., 2021; Rounce et al., 2023; Cook et al., 2023); see supp. Tab. S2 for an overview of data sets used).



For a second set of simulations, we adapted the model setup, calibration and validation procedure of OGGM default to utilize the higher resolution regional glacier change observations available for our study area (see previous sections). We refer to the adapted model as OGGM regional in the following.

To compare projections under different temperature scenarios, we used the median projected volume in 2100 in the +1.5°C scenario as a reference point (see Sect. 2.4.1). We computed the year in which this volume is reached under warmer scenarios as a metric to assess the speed of glacier disappearance in the ROI. Once the projected volume of individual glaciers reaches 0 km$^3$, it is not allowed to regrow (similar to the approach in Rounce et al., 2023). When counting how many glaciers remain in the ROI in a given year, we count only glaciers with a volume greater than 0 km$^3$ and apply no further volume or area 215 thresholds.

### 2.4.1 Climate input

OGGM uses historical climate data for calibration and initialisation and relies on bias-corrected global climate models for projections. The high-resolution regional climate dataset SPARTACUS was used to drive OGGM regional during the calibration period. SPARTACUS is a 1 km gridded product available from 1969 to present derived from meteorological observations (Hiebl 220 and Frei, 2015, 2017) (https://data.hub.geosphere.at/dataset/spartacus-v2-1m-1km). The glacier-scale values were computed by taking the mean of all SPARTACUS grid points within the respective glacier outlines and a 500 m buffer. The grid point closest to the outline's centroid was used when no grid points inside the glacier area were available.

To address potential biases in SPARTACUS due to limited observational data in complex terrain (Hiebl and Frei, 2017), we used a precipitation adjustment method detailed in Schuster et al. (2023a). For glaciers with winter mass balance observations, 225 we adjust the precipitation factor (pf) to match the observed mean winter mass balance. We obtain a pf of 1.8, 3, 2.2, and 2.7 for Vernagtferner, Hintereisferner, Goldbergkees, and Jamtalferner, respectively. For consistency across the study region, we used a constant value of 2.4 to correct SPARTACUS precipitation (the mean of all four glaciers). To assess the impact of precipitation corrections on our results, we conducted a sensitivity analysis using a pf of 1, 2, and 3 (see Sect. 4.2 and supp. Fig. S10).

To project glacier changes until 2100, we ran 47 climate scenarios consisting of 12 climate models together with different shared socioeconomic pathways (SSPs) from the coupled model intercomparison project 6 (CMIP6, Eyring et al. (2016)). The temperatures of the climate scenarios where bias corrected to SPARTACUS for the period 1961 - 1990. Our simulations were sorted into four categories based on global temperature change expected between 2071 and 2100 compared to the preindustrial era (1850 - 1900): +1.5°C, +2°C, +3°C, and +4°C, following a similar methodology as outlined in Rounce et al. (2023), using 235 +0.63°C temperature increase for 1850 - 1900 to 1986 - 2005 (IPPC, 2019). See supplementary Table S4 for an overview of model realizations and temperature bins. As the increase in global temperature does not directly reflect local conditions at glacier sites, we also calculated the area-weighted temperature signal (using 2017/18 areas). We found local temperature increases of +2.1°C, +2.8°C, +3.9°C, and +5.0°C, respectively. This highlights the stronger warming expected in the Alps compared to global average warming levels.





### 2.4.2 Model setup and validation

OGGM employs a novel dynamic calibration and initialisation workflow, which iteratively adjusts three model parameters to match three observation datasets (see Appendix A and Aguayo et al. (2023) for details). Each iteration dynamically updates one parameter to align the model more closely with observed data: the deformation-sliding parameter is adjusted to match glacier volume, the temperature index melt factor is adjusted to match geodetic mass balance, and the initial glacier state in the past is adjusted to match a glacier area at a specific timestamp. This calibration and initialization approach is applied to each glacier individually. OGGM default utilizes global datasets for calibration and initialization (using one glacier outline), while OGGM regional uses regional data and two glacier outlines at different timestamps.

For OGGM regional, we initialized the elevation band flowlines using the 2017/18 glacier outlines, the 2017/18 DEM (Tab. 2) and the volume estimates for 2017/18, i.e. the 2006 estimated volume minus the volume lost between 2006 and 2017 (section 2.3). During the dynamic initialization phase, our goal was to match the model output with the 2006 glacier area (GI3, Tab. 1) and the observed geodetic mass balance from 2006 to 2017/18. The geodetic mass balance was computed from observed volume change (section 2.2) and an assumed bulk density of 850 kg/m³ (following Huss (2013), further information in Appendix A). The starting year for the dynamic spin up of OGGM regional was set to 1979 as in OGGM default.

We extended the approach developed by Aguayo et al. (2023) by using two outlines instead of one. This enabled us to calibrate the cross-section angle of the elevation band flowlines by matching to the observed area change between the two outlines. Based on a sensitivity study for the whole region (see section 4.2 and supp. Fig. S9), we found that an angle of 27° is most suitable (setting $\lambda$ from 2 to 4, as defined in supp. Fig. S2), which is lower than the value of 45° used by OGGM default and other studies (e.g. Huss and Farinotti, 2012; Huss and Hock, 2015; Werder et al., 2019; Zekollari et al., 2019) and implies larger glacier area changes per ice thickness change and a stronger mass-balance elevation feedback than OGGM default.

For model validation, we relied on observational data not included in the dynamic calibration and initialization process, i.e. glacier area in 1997 (GI2, Tab. 1), total volume change from 1997 to 2006 (section 2.2), and glaciological mass balance measurements from three WGMS reference glaciers within our ROI (WGMS, 2023). Finally, we compared the projections by OGGM regional to those by OGGM default for all glaciers within our ROI and to results from five additional regional glacier modeling studies (Hanzer et al., 2018; Zekollari et al., 2019; Compagno et al., 2021; Rounce et al., 2023; Cook et al., 2023).

## 3 Results

### 3.1 Glacier volume and area change between 2006 and 2017/18

Between 2006 and 2017/18, 34.8 ± 6.1 km² of glacier area and 1.88 ± 0.05km³ of volume were lost in the study region (Tab. 5). The changes are equivalent to 19% of the 2006 area and roughly 23% of the 2006 volume, representing a mean annual volume loss rate of about 2%. Over 20% of glaciers in the study area lost more than half their area and volume since 2006 (69 lost more than 50% of the 2006 area, 71 lost more than half the 2006 volume). 9 glaciers lost more than 80% of their 2006 area, and 17 more than 80% of their volume.





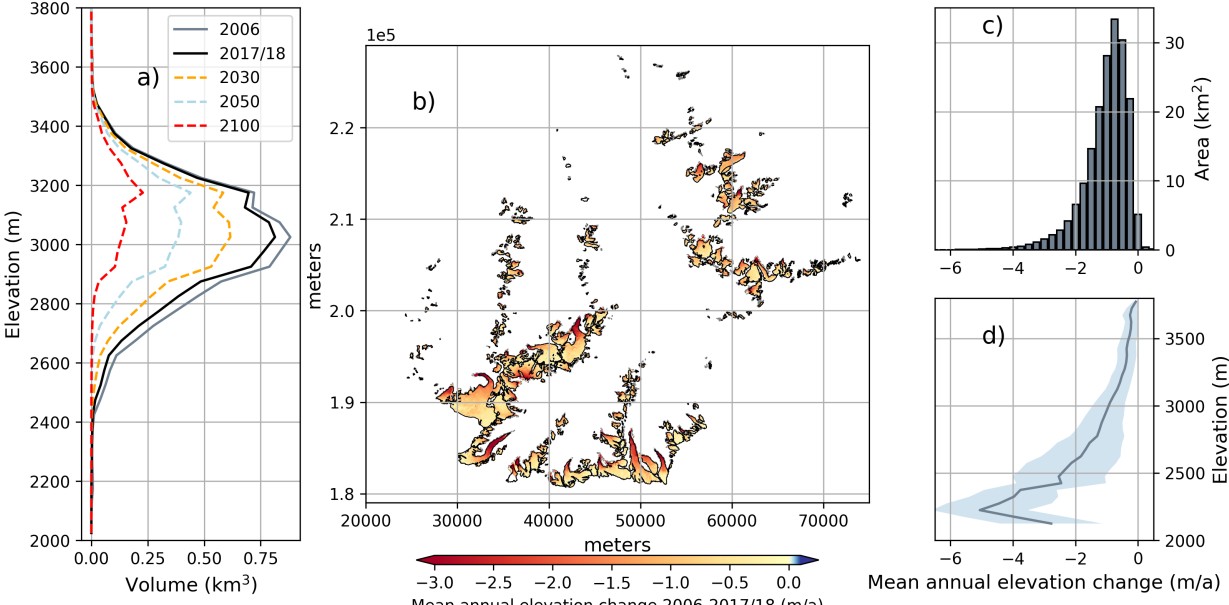

**Figure 3.** a) Glacier volume in the ROI per 50m elevation bin for 2006, 2017/18 and extrapolated to 2030, 2050, and 2100 for the constant change rate scenario based on 2006–2017/18 change rates. b) Mean annual elevation change (m yr$^{-1}$) for 2006–2017/18 in the study region within the 2006 glacier outlines, grid in meters, EPSG:31254 c) Histogram of mean annual elevation change, pixel counts converted to area in km$^2$. d): Mean annual elevation change per 50m elevation bands. Blue shading represents $\pm 1$ standard deviation.

Five glaciers that were classified as such in the 2006 inventory were no longer included in the updated 2017/18 inventory. All five were smaller than 0.1km$^2$ in 2006 (Tab. 6). Two small glaciers ($\sim$0.1 km$^2$) in the Stubai region, Glätte Ferner and Westlicher Grübl Ferner E., have slightly positive volume change and negative area change for the 2006–2017 period. Both are located in and/or directly below steep rock faces and the positive volume change is likely due to snow accumulation from avalanches.

Figure 3 shows mean annual elevation change mapped over the study region. On average, change was negative across the entire altitudinal range and only a very small fraction of the glacier area in the study region showed slight elevation gains. The mean elevation change rate for 2006–2017/18 is -0.92 m yr$^{-1}$. The mean loss rate has increased slightly compared to the previous inventory period 1997–2006 and is marginally lower than the -0.96 m yr$^{-1}$ found by Hugonnet et al. (2021a) for 2000-2020 (supp. Tab. S1).

Grouping glaciers by size, the magnitude of the elevation change rate in 2006–2017/18 is lowest for very small glaciers (-0.7 m yr$^{-1}$). However, very small glaciers have on average lost a greater percentage of their volume than other size classes, since the total volume is small (Tab. 5). At the scale of individual glaciers in the "small" (0.5-1 km$^2$) and "very small" (<0.5 km$^2$) size classes, the variability of percentage volume loss per glacier is high and the spread of values tends to increase with decreasing glacier size (Fig. 4 c, d). The fraction of glaciers that are classified as "very small" increases as glaciers shrink in



**Table 5.** Area and volume changes 2006–2017/18 for the entire study region and by glacier size class.

|  | total | < 0.5 km$^2$ | 0.5-1 km$^2$ | 1-5 km$^2$ | 5-10 km$^2$ | >= 10 km$^2$ |
|---|---|---|---|---|---|---|
| Nr. of glaciers | **317** | 264 | 21 | 25 | 6 | 1 |
| Change since 2006 | **-5** | +8 | -7 | -5 | -1 | 0 |
| Area 2017/18 in km$^2$ | **151.81±3.33** | 30.21±1.51 | 14.99±0.22 | 50.13±0.75 | 40.9±0.61 | 15.52±0.23 |
| Percentage of total area | **100%** | 20% | 10% | 33% | 27% | 10% |
| Area change since 2006, km$^2$ | **-34.8±6.1** | -6.6±2.4 | -4.1±1.2 | -10.8±1.2 | -12.2±1.0 | -1.1±0.3 |
| Volume change since 2006, km$^3$ | **-1.88±0.05** | -0.28±0.01 | -0.17±0.01 | -0.63±0.01 | -0.63±0.01 | -0.17±0.003 |
| Volume change as % of 2006 volume | **-22.8** | -35.2 | -28.9 | -23.5 | -22.5 | -12.3 |
| 2006–2017/18: Mean rate of elevation change (m yr$^{-1}$) | **-0.92±0.01** | -0.70±0.01 | -0.82±0.01 | -0.94±0.01 | -1.07±0.01 | -0.94±0.01 |

the study region. In 2017/18, 264 of 317 glaciers (83%) in the study area were very small (Tab. 5). The number of glaciers in the larger size classes has accordingly decreased slightly since 2006, except for the >=10 km$^2$ class, which consists of a single glacier (Gepatschferner, 15.5 km$^2$ in 2017).

Very small glaciers contribute about 40% of the total 2006–2017/18 area loss and about 20% of the volume loss. Conversely, glaciers larger than 5 km$^2$ (i.e., seven glaciers as of 2017/18) contribute about 20% of area loss and 40% of volume loss (Fig. 4 d). About half of the total area loss occurred in the northern sector (NW-NE), which is where most of the remaining ice and most individual glaciers are located (Fig. 4, a, b).

In terms of both area and volume, most of the ice in the study area is located in an elevation range from about 2900 to 3200m
(Fig. 3, a). Considering a "constant change rate" future scenario and the 2006 ice volume estimate, the majority of ice loss until 2030 will also take place in this elevation range. Beyond 2030, extrapolating the 2006–2017/18 change rates indicates that losses will be spread more uniformly throughout the elevation ranges that still contain ice (Fig. 3, a). This implies that areas of thinner ice in the elevation range of the 2017/18 maximum ice volume will have disappeared by 2030 and, hence, no longer contribute to ice loss. At the scale of individual glaciers, the "constant change rate" scenario suggests that almost half of the
glaciers in the study region will disappear by 2050. The greatest reduction in numbers occurs by 2030 at very small glaciers. By 2050, only two glaciers larger than 5 km$^2$ would be left in this scenario and almost all ice below 2800m would be gone.

### 3.2 Glacier evolution model performance

OGGM regional successfully simulates 299 of the 317 glaciers in the ROI, representing 99.2% of the total glacier area (see Appendix A). Focusing on glacier volume and area (Fig. 5, a, b), OGGM regional and OGGM default runs match the respective
data used for calibration, showing that the iterative dynamic calibration and initialization work as expected. Additionally,



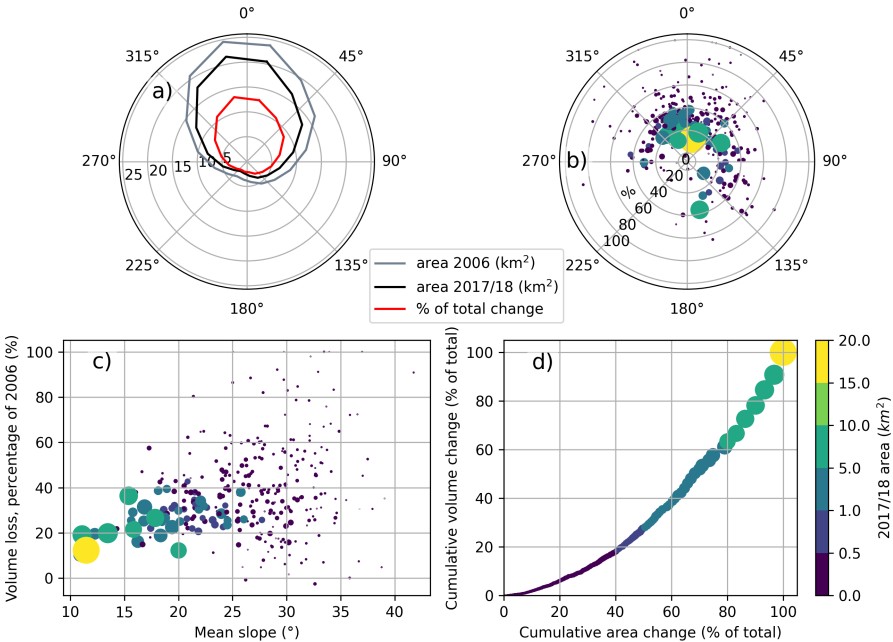

**Figure 4.** a) Area distribution by aspect for 2006 and 2017/18, and percentage of total area change. b) Mean aspect of glaciers in the study region (polar axis) and 2006–2017/18 volume change as percentage of the 2006 volume (radial axis). c) Mean slope of glaciers in the study region (horizontal axis) and 2006–2017/18 volume change in % rel. to the 2006 volume (vertical axis). d) Relative, cumulative contribution to area and volume change (2006–2017/18) sorted by size of individual glaciers. In b), c), and d) marker size is scaled to glacier sizes and color indicates size classes.

**Table 6.** Glaciers that disappeared between the 2006 and 2017/18 inventories and are no longer included in the updated inventory.

| Name (Austrian inventory number) | Area (km$^2$) in 2006 |
|---|---|
| Wurmkogel Ferner (2054) | $0.036 \pm 5\%$ |
| Plattenkogel Ferner (2053) | $0.013 \pm 5\%$ |
| Innerer Pirchelkar Ferner E (2162) | $0.020 \pm 5\%$ |
| Südlicher Petzner Ferner (2146) | $0.034 \pm 5\%$ |
| Nördlicher Petzner Ferner (2147) | $0.005 \pm 5\%$ |

OGGM regional agrees with the regional validation data that was not used during calibration, i.e. 1997–2006 volume change and 1997 area (Fig. 5 a, b).

While trends in volume change are similar for OGGM regional and OGGM default, the OGGM regional produces a substantially faster area decline compared to OGGM default during the period 1979-2020. This is a combined effect of the adapted





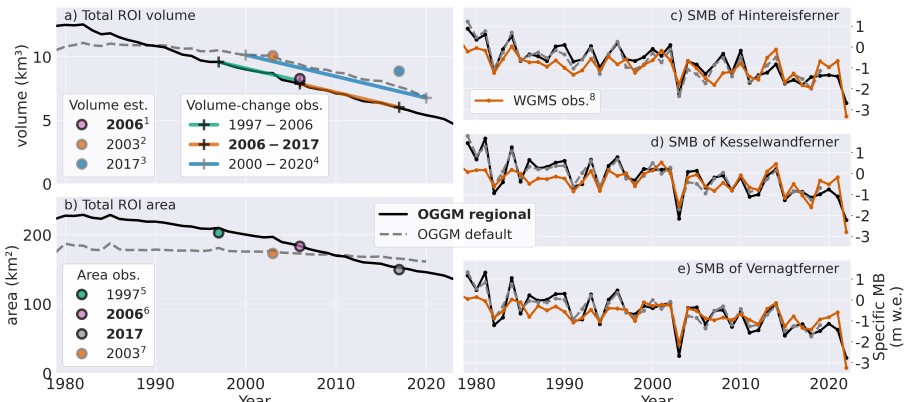

**Figure 5.** Comparison between OGGM regional and OGGM default against observations across the total ROI for glacier volume (a) and area (b), as well as the specific mass balance (SMB) for Hintereisferner (c), Kesselwandferner (d), and Vernagtferner (e). In panels a and b, observations that informed the calibration and initialisation of OGGM regional are highlighted in bold. OGGM default is calibrated with the volume 2003, the volume-change obs. 2000 - 2020 and the area 2003. Markers distinguish between regional (black edges) and global (gray edges) observations. Observations cited with a superscript originate from external studies, while those without come from this study. Volume estimate sources are [1] Helfricht et al. (2019a), [2] Farinotti et al. (2019), and [3] Millan et al. (2022). The volume-change observation is from [4] Hugonnet et al. (2021b). Area measurements are taken from [5] GI2 Kuhn et al. (2013), [6] GI3 Fischer et al. (2015a), and [7] Randolph Glacier Inventory Consortium (2017). Specific mass balance data are sourced from [7] WGMS (2023). A detailed comparison can be found in the supplement (supplementary Fig. S6 and supplementary Table 3S).

cross-section angle (see section 2.4.2) and the use of a more recent outline during the glacier bed inversion (more available information about the bedrock). Both OGGM regional and OGGM default accurately replicate observed mass balance trends and yearly variability, especially from 2000 onwards for three reference glaciers in the ROI (Fig. 5, c, d, e and Tab. A1).

## 3.3 Model projections until 2100

The aggregated model outcomes for four temperature scenarios (+1.5°C, +2°C, +3°C, and +4°C above pre industrial levels)
are shown in Figure 6 (glacier volume) and supplementary Figure S7 (glacier area). The strong ice loss of the past decades will continue for all scenarios, with little differences between scenarios until about 2035.

Post-2035, the +1.5°C global temperature scenario predicts a more gradual decline, suggesting that glacier evolution will stabilize towards the end of the century with around 2.7% of the 2017 volume remaining until beyond 2100, most of it above 3000 m. The +2°C scenario diverges from the warmer scenarios around 2040 and forecasts 0.4% of the 2017 volume remaining
320 in 2100. Decrease rates under the +3°C and +4°C scenarios are similar because deglaciation in the ROI is largely complete before the temperatures of those scenarios deviate.

Defining the projected glacier volume for the +1.5°C scenario in 2100 (0.18 km³, 2.7% of the 2017 volume) as a baseline, we find that this volume is reached around 30 years earlier for the +2°C scenario, in 2071 [5th and 95th percentile: 2055 and





beyond 2100]. For higher temperature scenarios, this reference volume is reached even earlier: in 2063 [2055 to 2079] for +3°C
and in 2060 [2050 to 2075] for +4°C. For 2050, we project a glacier volume of 1.36 km$^3$ (20.7% of the 2017 volume) in the
+1.5°C scenario. For the +2°C and +3°C scenarios, the projected volume in 2050 decreases to 0.98 km$^3$ (14.9%) and 0.74 km$^3$
(11.2%), respectively. This reflects the fact that stringent mitigation measures are needed to slow down warming before 2050.

The number of individual glaciers in the ROI decreases faster in scenarios with a greater temperature increase. How many
glaciers remain by 2100 depends strongly on the scenario (Fig. 7 and supp. Fig. S4). Considering the model run closest to
the median volume evolution, as well as the model runs closest to the 5th and 95th volume percentile in brackets, we find
71 [29 to 126] glaciers in 2100 for +1.5°C, 17 [16 to 82] glaciers for +2.0°C, 8 [5 to 17] glaciers for +3.0°C and 2 [2 to 9]
glaciers for +4.0°C remaining in 2100. The main portion of the remaining ice volume in all cases will be concentrated above
3200 m. We note that we count any remaining ice within the glacier boundary as a "glacier" regardless of area or volume. The
disappearance of very small glaciers contributes most strongly to the reduction in the overall number of glaciers in the ROI.
Notably, about 100 of the glaciers in the 2017/18 inventory are projected to disappear by 2030 even in the +1.5°C scenario.

Figure 8 and supplementary Figure S3 show projected changes for specific glaciers in the ROI based on median model
predictions (model closest to the volume median of the full scenario ensemble). The figures don't show the full range of
possible outcomes for each scenario and are intended as a general overview of the possible future evolution of the example
glaciers. Differences between the +1.5°C and the +2°C scenario are small until 2050, similar to Fig. 6. As time goes on, the
speed at which each glacier retreats starts to vary, with almost complete glacier loss in 2100 for +2°C. The remaining end of
century ice for +1.5°C is mainly located where glaciers can retreat to higher elevations, as is the case at Hintereisferner (HEF),
Vernagtferner (VF), Gepatschferner (GPF) or Taschachferner (TF). Keselwandferner (KWF), is projected to disappear even
under +1.5°C. Notably the region shown in Fig. 8 contains two thirds of the total remaining volume in 2100 for +1.5°C (0.17
km$^3$ compared to the total ROI volume of 0.24 km$^3$). Compared to the +1.5°C and +2°C scenarios, glacier retreat is faster for
the +3°C and +4°C scenarios and differences between the warmer scenarios are already apparent by 2050 (supp. Fig. S3).

## 4 Discussion

### 4.1 Challenges of observing rapid glacier loss in the ROI

The high resolution observational data sets of area and volume change provide a detailed record of glacier change in the ROI
over the last decades and allow for improved calibration and validation of regional and global modeling studies. As glacier loss
progresses, the relative contribution of small and very small glaciers to total glacier area and volume increases. Hence, accurate
monitoring of the evolution of such features is essential to obtain a complete picture of the state of the regional cryosphere.
Medium resolution satellite imagery commonly used for larger scale glacier inventories is typically considered appropriate for
mapping glaciers larger than 0.01 km$^2$ (Paul et al., 2009) but has limitations for smaller features and debris covered areas. In
the ROI, 9% of glaciers were smaller than 0.01 km$^2$ and about 80% were smaller than 0.5 km$^2$ as of 2017/18. High resolution
topographic data that include information on surface elevation change aids outline detection and is particularly important for
confident mapping of small glaciers with changing debris cover, which are becoming increasingly common (e.g. Fischer et al.,





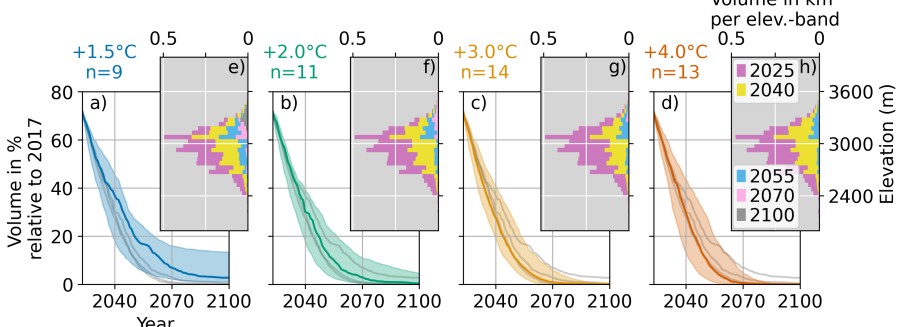

**Figure 6.** Median (colored lines) and 5th and 95th percentile (shading) of the OGGM regional projections per future global temperature scenario as percentage of 2017 glacier volume in the ROI (a, b, c and d) from 2023 to 2100. The grey lines in each subplot show the median of the other three scenarios for reference. Temperature increase and number of climate model realizations n per scenario stated above the volume evolution plots. Insets (e, f, g and h) show the distribution of ice volume per 50 m elevation bands in different years for the four temperature scenarios, for the model run closest to the volume median of the scenario ensemble (see supp. Tab. S4).

2014, 2021). Since uncertainties in glacier area dominate the overall uncertainty in observed glacier mass change in regions with many relatively small glaciers (Hugonnet et al., 2021a), improved area observations are a key component to reducing uncertainties in further analyses.

Processes relevant to local glacier elevation change can also be resolved in greater detail as the spatial resolution and quality of the topographic information used to derive surface change increases. In the ROI, two very small glaciers show a volume increase and an area decrease between 2006 and 2017/18. Closer inspection shows that the volume increase occurred exclusively in or near steep cliff faces and positive surface elevation change is likely due to snow accumulation from avalanches. This exemplifies that snow redistribution and other topography driven processes can play a key role for local glacier mass balance

especially at very small features, as has been found in previous studies from the Alps and other mountain regions (e.g. Debeer and Sharp, 2007; Abermann et al., 2011; Fischer et al., 2015c; Menounos et al., 2019; Florentine et al., 2020).

  As deglaciation progresses, processes such as basal melt and albedo change related to loss of firn and darkening of glacier surfaces can create feedback loops that accelerate melt. On the other hand, increasing debris cover can potentially slow the melt of certain features. The contributions of melt-accelerating feedbacks to overall mass loss are generally captured in high

resolution DEM differencing if they occur at the glacier surface. However, basal melt and associated thinning of the ice does not result in surface elevation change visible in DEMs and, hence, is not apparent in DEM differencing until the affected section of ice disintegrates or forms "collapse features" (e.g. Stocker-Waldhuber et al., 2017; Kellerer-Pirklbauer and Kulmer, 2019; Egli et al., 2021; Wytiahlowsky et al., 2024), leading to an underestimation of mass loss in the respective observational data. To accurately track regional glacier evolution under current conditions of rapid change, increased frequency of inventorization

with high resolution topographic data that also allows for the assessment of volume change under debris cover is key.





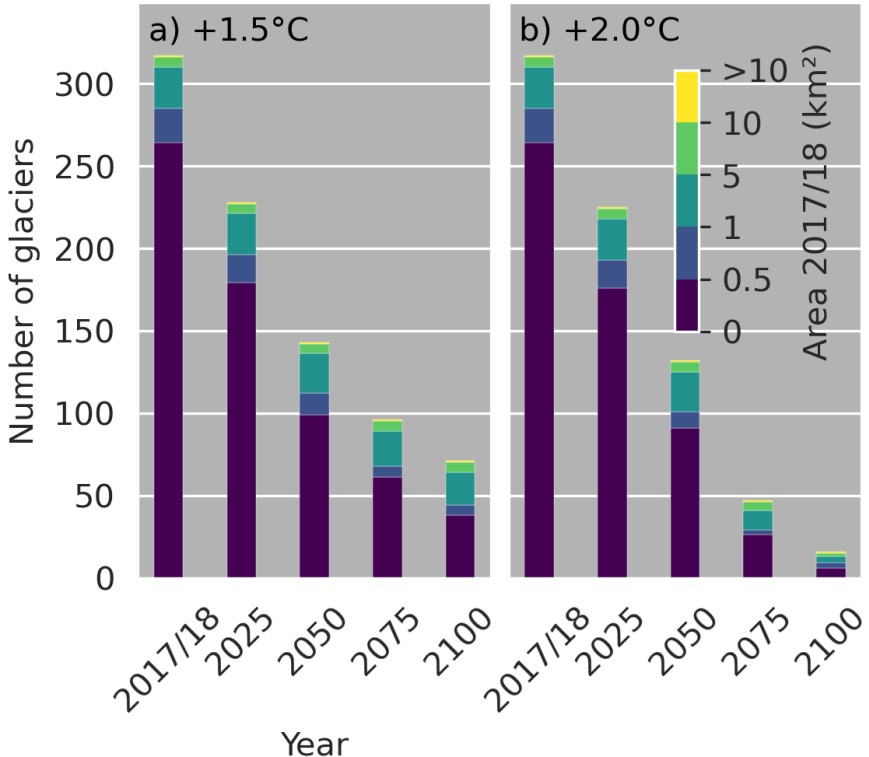

**Figure 7.** OGGM regional projected future changes in the number of glaciers under +1.5°C (a) and +2.0°C (b) scenarios, illustrated by single models which are closest to the ensemble median volume (see supp. Tab. S4). Colors represent area size classes in 2017/18 (see also Tab. 5).

## 4.2 OGGM sensitivity and uncertainty analysis

During the setup of OGGM regional (see section 2.4 and related subsections), we defined fixed values for the precipitation factor, for the bulk density for converting the observed volume change into a mass change, and for the trapezoidal wall angle of the glacier bed defined through $\lambda$ (see supp. Fig. 2). We conducted sensitivity studies by varying these three parameters to assess their impact on calibration and future projections, see supplementary Figures 8, 9, and 10.

Precipitation factors of 2, 2.4, and 3 all have similar performances during calibration. However, With a precipitation factor of 1, we observed a large mismatch between model results and the observed area 1979 and volume before 2017/18 (see supp. Fig. 10) compared to factors of 2, 2.4, and 3. This occurs because the dynamic spinup is unable to match the 2006 area without using unrealistically large 1979 glacier states to account for insufficient accumulation. In such cases, the model falls back to 385     using a fixed geometry spinup, which results in poor matching of past observations (see Appendix A).

    The model's sensitivity to the bulk density factor and trapezoidal-$\lambda$ is seen in different volume and area trends. A higher bulk density results in larger volumes and areas in 1980, but smaller ones by 2023, compared to a lower density (see supp.



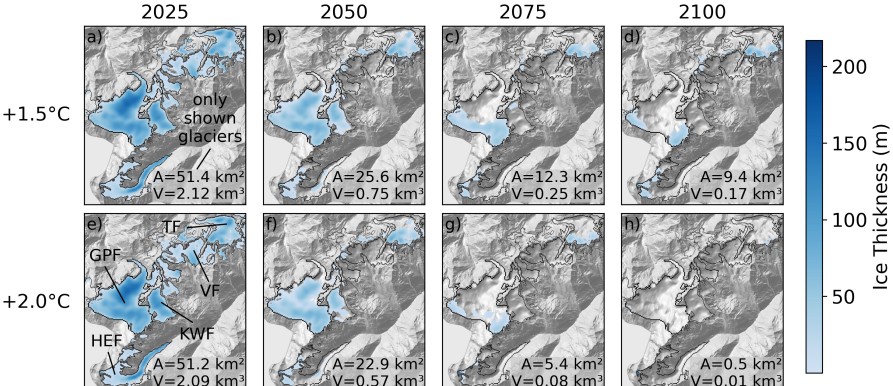

**Figure 8.** OGGM regional projected ice distribution across 27 glaciers in a subset of the ROI, including Hintereisferner (HEF), Kessel-wandferner (KWF), Vernagtferner (VF), Taschachferner (TF), and Gepatschferner (GPF) for a +1.5°C global warming scenario (a, b, c, and d) and a +2.0°C warming scenario (e, f, g, and h). The figure shows single models which are closest to the ensemble median volume (see supplementary Table S4). For each subfigure, the total area (A) and the total volume (V) of the 27 glaciers are noted in the bottom right corner. Shades of blue represent the ice thickness at each grid point. The 2017 outlines are shown in black.

Fig. 8, a, f). This causes the curves to tilt around the 2006 observations (as those are matched during the dynamic spinup, see section 2.4.2). Similarly, a larger trapezoidal-$\lambda$ increases the area in 1980 and decreases it by 2023, but has little impact on volume. Therefore, trapezoidal-$\lambda$ is a useful parameter for matching observed area changes alongside volume changes during calibration. Our default values of 850 kg m$^{-3}$ for bulk density (Huss, 2013) and 4 for trapezoidal-$\lambda$ show the best simultaneous match to the observed area and volume change during the calibration period (2006 to 2017/18). More past observations of both area and volume are needed to better constrain these parameters.

In terms of projections, the biggest impact on the timing of crossing the reference volume of 0.18 km$^3$ (see section 3.3) comes from using a precipitation factor of 1. In scenarios with +2°C, +3°C, and +4°C warming, the reference volume is reached 7, 9, and 8 years later, respectively, compared to the default factor of 2.4. The other precipitation factors result in smaller changes, with a maximum delay of 2 years. Adjusting the bulk density factor and trapezoidal-$\lambda$ shifts the year by a maximum of 2 and 3 years, respectively.

When considering the range of OGGM regional projections for different temperature levels, we see that the timing of crossing the reference volume varies by about ± 10 to 15 years (based on the 5th and 95th percentiles). This is the same order of magnitude as for a precipitation factor of 1, emphasizing the critical role of accurate precipitation data for model calibration. However, due to the glacier's high altitude, complex terrain, limited observations, and small-scale topographic effects, precipitation remains one of the most difficult variables to measure at the glacier level.



### 4.3 Comparison with previous studies

OGGM regional predicts the most rapid glacier decline in the ROI compared with global studies by Zekollari et al. (2019), Compagno et al. (2021), Cook et al. (2023), and Rounce et al. (2023), and OGGM default (Schuster et al., 2023a). Specifically, the median year of all projections when the reference point of 0.18 km$^3$ (+1.5°C end of century volume of OGGM regional) is crossed is 2066. OGGM default reaches the reference point approximately 9 years later. This difference underscores the influence of high resolution regional calibration data on model projections, given that both models otherwise employ the same 410 approach (Fig. 9, Tab. 7). The enhanced temporal coverage of observational input data available for OGGM regional allows for additional validation and increases confidence in its results compared to OGGM default.

The closest projections to OGGM (default and regional) are from Rounce et al. (2023, see Fig. 9 and Tab. 7), who used the same calibration data (Hugonnet et al., 2021a) and climate forcing (CMIP6) as OGGM and coupled OGGM dynamics with the PyGEM mass balance model (Rounce et al., 2020a, b). Differences are larger with the results of Zekollari et al. (2019) 415 and Compagno et al. (2021), who used the GloGEMflow model (Huss and Hock, 2015; Zekollari et al., 2019) forced with CORDEX and CMIP6 climate data, respectively. Zekollari et al. (2019)'s projections indicate a relatively large amount of ice remaining in the ROI through 2075 and are closest to the linear extrapolation of 2006–2017 observations. Compagno et al. (2021) limited future warming to +1°C to +2°C degrees in their projections. Their glacier evolution trajectory is comparable to OGGM default and Rounce et al. (2023) until about mid-century and diverges to a more glacier-favorable outlook thereafter, 420 which is to be expected when only running cooler scenarios as in Compagno et al. (2021) (+1°C to +2°C) and comparing it to warmer scenarios for OGGM and Rounce et al. (2023) (+1.5°C to +4.0°C) (Fig. 9, Tab. 7). It is not possible to further disentangle the reasons for these discrepancies, but the differences in climate forcing, model calibration, model initialization and mass-balance modeling choices are probably responsible.

Cook et al. (2023) used the Instructed Glacier Model to project glacier evolution until 2050 and find a higher remaining 425 volume of 4.56 km$^3$ of glacier ice compared to the other studies (around five times larger than the all-scenarios median volume of OGGM regional). This is primarily due to their initialization strategy, which starts the model run in 2020 with the glacier outline dates of the RGI6 (i.e. 2003) without a prior dynamic run to account for glacier evolution since 2003 (OGGM regional: -38% volume and -25% area change from 2003 to 2020).

Aside from OGGM regional, Hanzer et al. (2018) is the only modeling study focused on the region of interest (ROI), 430 specifically on the Ötztal (excluding Pitztal and Stubaital). They used parts of the AMUNDSEN hydroclimatological model (Strasser et al., 2024) to define glacier mass balance, in particular the precipitation output (including snow aging) and an energy balance model for estimating snow and ice melt. Glacier dynamics were modeled using the $\Delta h$ method of Huss et al. (2010), and projections relied on EURO-CORDEX climate data (similar to Zekollari et al. (2019)). Comparing the median volume of Hanzer et al. (2018)'s simulations with OGGM regional for Ötztal only, they project an end-of-century volume of 0.27 km$^3$, 435 while OGGM regional projects 0.01 km$^3$. OGGM regional shows a faster decline, reaching Hanzer et al. (2018)'s projected end-of-century volume by 2061 (5th and 95th precentile: 2050 to beyond 2100). The difference between the studies is hard to pinpoint due to the differing approaches: Hanzer et al. (2018) uses a more physically detailed energy balance model requiring





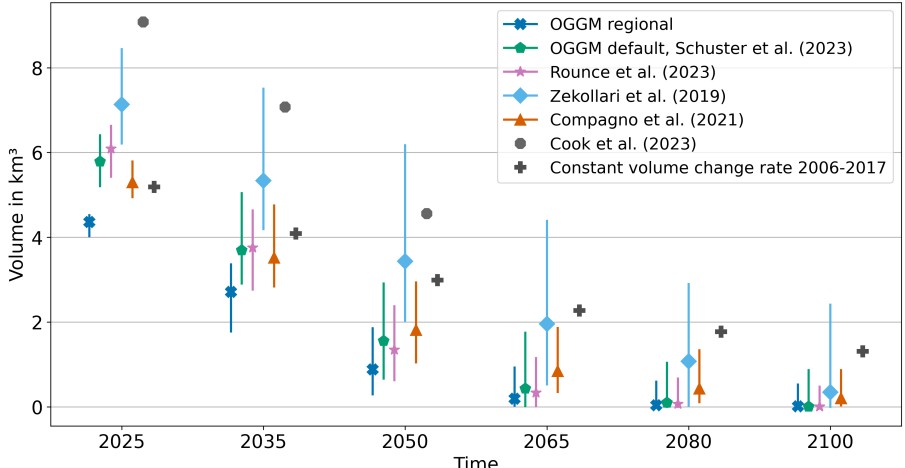

**Figure 9.** Projected total volume in km$^3$ of our ROI for the years 2025, 2035, 2050, 2065, 2080 and 2100 for OGGM regional, OGGM default (Schuster et al., 2023a), Rounce et al. (2023), Zekollari et al. (2019), Compagno et al. (2021), Cook et al. (2023), and a constant volume change rate 2006–2017. Shown are the median values (markers) with 5th and 95th percentile (line) for all conducted experiments of the individual studies, if more than one is available. For Cook et al. (2023), only the realisation using a linearly interpolated observed mass balance trend is included.

higher-resolution meteorological data and a $\Delta h$ parametrisation for glacier dynamics, while OGGM regional relies on a simpler temperature index model together with flowline dynamics. Further, both models rely on different datasets for calibration.

Overall, while there are variations in the projections depending on the modeling approach and climate scenarios, all studies agree that we will see a substantial loss of glacier ice in the ROI in all warming scenarios. However, the speed and magnitude depends on the model and scenario, with the fastest and largest losses projected by OGGM regional in this study. This highlights the importance of region-specific high resolution observations for model calibration, but also of model validation for increasing the confidence in the model (OGGM regional best resamples past observations as shown in the supplementary material (sup.

Fig. S6 and sup. Tab. S3). Refined regional projections could offer critical insights for planning climate adaptation strategies, like estimating future glacier runoff as a water resource for ecosystems, irrigation or hydropower production.

### 4.4 Improving future modeling studies

#### 4.4.1 Ice volume estimates

Absolute past ice volume serves as a starting point for future projections and is crucial for model initialization. Discrepancies in

the starting volume between different studies clearly impact the estimated time to deglaciation in the ROI (Fig. 9 ). Accordingly, improved estimates of past and current ice volume would improve confidence in projections. Ice volume estimates often use the apparent mass balance gradient to account for surface mass balance distribution and ice flux (Huss and Farinotti, 2012; Farinotti et al., 2019; Helfricht et al., 2019a), yet ice flux is limited at small, rapidly receding glaciers as found in our study





**Table 7.** This table compares the total ROI volume evolutions of OGGM regional, OGGM default (Schuster et al., 2023a), Rounce et al. (2023); Zekollari et al. (2019); Compagno et al. (2021); Cook et al. (2023) and a constant volume change rate 2006–2017 with the median and the 5th and 95th percentiles of all conducted experiments for the years 2025, 2050, 2075 and 2100. Further, it shows the year when the reference point of 0.18 km3 is crossed (see section 3.3) for all scenarios and all models. The crossing of the reference point for different temperature levels is shown for OGGM regional, OGGM default (Schuster et al., 2023a), and Rounce et al. (2023).

| | OGGM regional | OGGM default | Rounce et al. (2023) | Zekollari et al. (2019) | Compagno et al. (2021) | Constant volume change rate 2006–2017 | Cook et al. (2023) |
|---|---|---|---|---|---|---|---|
| Volume all realisations in km³ | | | | | | | |
| 2025 | 4.36 [4.0 to 4.5] | 5.8 [5.2 to 6.4] | 6.1 [5.4 to 6.6] | 7.1 [6.2 to 8.4] | 5.3 [5.0 to 5.8] | 5.2 | 9.1 |
| 2050 | 0.88 [0.3 to 1.9] | 1.6 [0.7 to 2.9] | 1.4 [0.6 to 2.4] | 3.4 [2.0 to 6.2] | 1.8 [1.1 to 2.9] | 3.0 | 4.6 |
| 2075 | 0.1 [0.0 to 0.7] | 0.2 [0.0 to 1.3] | 0.1 [0.0 to 0.8] | 1.2 [0.1 to 3.2] | 0.5 [0.1 to 1.4] | 1.9 | — |
| 2100 | 0.0 [0.0 to 0.5] | 0.0 [0.0 to 0.8] | 0.0 [0.0 to 0.5] | 0.3 [0.0 to 2.4] | 0.2 [0.0 to 0.9] | 1.3 | — |
| Year when reference point 0.18 km³ is crossed | | | | | | | |
| All realisations | 2066 [2054 to >2100] | 2075 [2059 to >2100] | 2072 [2058 to >2100] | >2100 [2073 to >2100] | >2100 [2073 to >2100] | >2100 | >2100 |
| +1.5°C | >2100 [2067 to >2100] | >2100 [2090 to >2100] | 2095 [2076 to >2100] | — | — | — | — |
| +2.0°C | 2071 [2055 to >2100] | 2079 [2072 to >2100] | 2073 [2067 to >2100] | — | — | — | — |
| +3.0°C | 2063 [2055 to 2080] | 2069 [2062 to 2084] | 2068 [2059 to 2078] | — | — | — | — |
| +4.0°C | 2060 [2051 to 2075] | 2069 [2061 to 2078] | 2065 [2060 to 2075] | — | — | — | — |

region. Helfricht et al. (2019a) found that the apparent mass balance gradient parameter needs to be well calibrated to match

regional observations, which implies that global ice thickness estimates likely struggle to accurately represent small and very small glaciers in a state of pronounced imbalance.

Improved availability of measured ice thickness would allow for better calibration of regional ice volume estimates. Uncertainties in glacier area and outlines also impact the ice volume estimates due to the relation between area and volume, so that, for example, a failure to separate ice complexes into individual glaciers in inventories introduces a bias to greater ice thickness

(Farinotti et al., 2019). Once more, this underlines the need for comprehensive glacier inventories based on high resolution data.





### 4.4.2 Unresolved processes and data assimilation

In general, the temperature index model and dynamics approach of OGGM are simplified representations of processes relevant for glacier evolution. Melt-accelerating feedback mechanisms are not resolved in the model, likely leading to an underestimation of projected melt. Incorporating albedo parameterizations calibrated with observations of glacier surface conditions as well as model implementation of subsurface processes would be desirable for future work.

The computationally cheap trapezoidal glacier geometry also does not resolve the reality of many glaciers. Calibrating a more geometrically complex model with the available data on past glacier states is challenging, as distributed mass balance and mass balance variability can be strongly impacted by small scale processes such as avalanches or wind drift. Nonetheless, including additional observational data helps to better constrain the trapezoidal glacier geometry, e.g. by providing additional information of the ice-free bedrock (using the 2017/18 outlines for bed inversion) or adapting the slope angle of the trapezoidal bed shape by matching observed area changes between two outlines (2006 to 2017/18). Comparing OGGM regional with OGGM default highlights the importance of model calibration on multiple data sets (Fig. 5).

The climate data used to drive the model is another key component of the model setup. Here, we find that the higher resolution regional reanalysis data (SPARTACUS) used in OGGM regional contributes to improved matching of past glacier change, although there is room for improvement regarding precipitation in mountainous terrain. The global climate model output used to force future glacier evolution is coarse in comparison and the spread in global climate models contributes substantially to the range of projected glacier evolution. To reduce the uncertainty in the projections stemming from the climate forcing data, high resolution regional climate modeling or downscaled global climate model output would be highly desirable.

Improving projections requires not only better observations and climate data but also models that can effectively integrate them. Flexible data assimilation methods are essential, as they enable the use of various observational data over time. Additionally, new approaches that account for data uncertainty and propagate it through the model to the projections are crucial. Projections should be accompanied by uncertainty estimates, clearly identifying the sources, whether from the model, observations, or climate projections. This information would guide future research and observational campaigns, focusing efforts on reducing the most significant uncertainties.

## 5 Conclusions

Glaciers in the region of interest (ROI) in the Ötztal and Stubai Alps are receding rapidly and expected to disappear before the end of the century if global warming is not limited to +1.5°C above pre-industrial temperatures. For the first time, we were able to simultaneously match observed area-changes together with observed volume-changes in a regional-scale glacier model. This allows for detailed assessments of individual glaciers' trajectories under different climate scenarios. Our results generally show a faster decline of glaciers in the ROI than comparable modeling studies. About a third of the over 300 glaciers listed in the 2017/18 inventory in the ROI is projected to disappear by 2030 in all scenarios. This includes mainly very small glaciers at relatively low elevations. Larger glaciers and ice in the highest elevations of the ROI are projected to persist longer and their evolution depends strongly on future warming. Between 2006 and 2017/18, roughly 35 km$^2$ of glacier area and 1.9



495   km$^3$ of glacier volume were lost at mean rates of -0.9 m elevation change per year. Glaciers in the ROI will disappear almost completely in the coming decades. Limiting warming to <+2.0°C results in delayed glacier loss compared to the +3°C scenario. If a low emission scenario with less than +1.5°C warming can be achieved, remnants of the current glacier volume (2.7% of the 2017 volume) could persist through the end of the century. The present warming trajectory of about +2.7°C results in near complete ice loss by about 2075 (<1% of the 2017 volume left). Remaining uncertainties originate from a combination

of factors difficult to evaluate or implement in the model, in particular uncertainties in near-term climate variability, and the presence of feedbacks currently unaccounted for in the model.

*Code and data availability.*   The 2017/18 glacier outlines for the ROI are available on the pangaea data platform at Helfricht et al. (2024a) (Ötztal) and Helfricht et al. (2024b) (Stubaital). The code of the OGGM version used for OGGM regional is available under Maussion et al. (2023) and the scripts used for conducting the OGGM regional model run as well as the model projections for the entire region are available

at Maussion and Schmitt (2024)

## Appendix A:  Dynamic calibration and initialisation workflow

For this study, a novel dynamic calibration and initialisation procedure was developed to match heterogenous observations (two area observations, one volume estimate, and one volume change observation) spread over the calibration period. The parameters adjusted during the transient calibration are the deformation-sliding parameter (to match glacier volume), the temperature index

melt factor (to match geodetic mass balance), and the initial glacier state in the past (to match one glacier area). The second area was used during inversion.

This is achieved using three nested loops, where each loop tries to minimize the difference between one single observed and modeled variable by adapting one model parameter. Since a change in one model parameter also influences the others, this is done iteratively. The final outcome is a consistent dynamic model run calibrated to all observations, which initializes

the OGGM projections with a best estimate of present day glacier state. A detailed description of the nested loops can be found in Aguayo et al. (2023) and in the OGGM documentation. In contrast to Aguayo et al. (2023) we used two different area estimates, one for the inversion and the other for dynamical matching during spinup.

To compare model results with geodetic observations, we convert the observed and modeled volume change $\Delta V (\mathrm{m}^{-3})$ into a geodetic mass balance $mb_{geo}$ (kg m$^{-2}$ yr$^{-1}$) using

$$mb_{geo} = \frac{\Delta V}{A * \Delta t} * f_{\Delta V}$$

where $A$ (m$^2$) is the area of the reference outline , $\Delta t$ (yr) is the length of the period over which $\Delta V$ was observed and $f_{\Delta V}$ (kg m$^{-3}$) is for converting volume into mass. Following commonly used practice (Hugonnet et al., 2021a; Rounce et al., 2023; Zekollari et al., 2024), we use a density of 900 kg m$^{-3}$ to convert OGGM volume change to mass, and 850 kg m$^{-3}$ to convert geodetic $\Delta V$ observations to mass (Huss, 2013; Hugonnet et al., 2021a). Density conversion for glacier mass change



assessment is an unsolved challenge (Berthier et al., 2023). Therefore we investigated the influence of this density conversion factor by conducting additional runs with 790 kg m$^{-3}$ and 900 kg m$^{-3}$ (see section 4.2 and supp. Fig. S8).

For each observation type we defined a range which we consider a satisfactory match, for volume $\pm 1\%$ of total glacier volume, for geodetic mass balance $\pm 1$ kg m$^{-2}yr^{-1}$ and for area $\pm 1\%$ of glacier area. Depending on whether the dynamic calibration matches the defined ranges there are three possible outcomes: all parameters match observations within provided uncertainty (full success); volume and area match but geodetic mass balance is incomplete (part success); volume matches but no past glacier state can be found that matches the area, resulting in a 'fixed geometry spinup', or 'static spinup'. In this case, a constant glacier surface geometry is assumed before the area observation date, and the area only evolves dynamically afterwards. Similarly, the volume is calculated backwards from the outline date (using a constant area), and only evolves dynamically afterwards. For OGGM regional, the dynamic spinup fully succeeds in simulating 76 glaciers (25.4%), covering 48.3 km$^2$ (32.3%). It partially succeeds for 173 glaciers (57.9%), covering 98.8 km$^2$(66%), and employs a 'fixed geometry spinup' approach for the remaining 50 glaciers (16.7%), which span 2.5 km$^2$ (1.6%) (see supp. Fig. S5).

We try to avoid the use of this 'fixed geometry spinup' as far as possible, therefore the dynamic spinup tries to shorten the spinup period for two special cases. The special cases refer to the matching of the target area. First, if the modeled glacier area is too small even when the glacier is close to growing outside the domain boundary. Second, if the modeled glacier area is too large even when we start from an ice free glacier surface. In the case of a shortened period we add a 'fixed geometry spinup' at the beginning to ensure a smooth evolution of total volume and area, with no jumps due to the changing number of glaciers. With this strategy OGGM regional dynamically initializes 171 glaciers (53.9%), covering 136.8 km$^2$ (90.7%), in 1979 and shortens the start year of the dynamic spinup to 1988 and 1996 for 39 glaciers (12.3%), covering 6 km$^2$ (4%) and 4.3 km$^2$ (2.9%) respectively (see supp. Fig. S5).

To showcase the advantages of the dynamic spinup over the static spinup we also conducted a run using the same regional observations as OGGM regional, but only using a static spinup. Especially when comparing the model performance in the past, we see that the static spinup performs worse in matching the observations used for validation (see supp. Fig. S11). Further, the dynamic spinup brings the modeled specific mass balance closer to the observed specific mass balance from WGMS in the period 2000 to 2020 (see Tab. A1). All this enhances our confidence that this new calibration and initialisation methodology provides significant added value.

*Author contributions.* LH and PS contributed equally to this work and share first authorship. PS developed the dynamic calibration strategy for OGGM regional and produced the future projections. LH carried out the volume change computations and generated the calibration data for OGGM regional. KH mapped the 2017/18 glacier outlines. LS provided advice on the method for defining the precipitation factor and computed the temperature warming for each climate scenario. FM and JA contributed to the analyses of model output and observations. LH, PS, and FM conceptualized the study and wrote the paper with input from all co-authors.



**Table A1.** WGMS Mean +/- standard deviation of specific mass balance in kg m$^{-2}$ for the years 2000 to 2020. Dynamic spinup, fixed geometry spinup of OGGM regional and OGGM default (Schuster et al., 2023a) mean bias and root mean squared error (RMSE) computed against WGMS.

| 2000 - 2020 WGMS MB in kg m$^{-2}$ | | Hintereisferner -1123 ± 557 | Kesselwandferner -474 ± 588 | Vernagtferner -829 ± 478 |
|---|---|---|---|---|
| OGGM regional Dynamic spinup | Mean bias | -66 | 50 | 134 |
| | RMSE | 514 | 343 | 328 |
| OGGM regional Fixed geometry spinup | Mean bias | 100 | 105 | 304 |
| | RMSE | 559 | 368 | 475 |
| OGGM default | Mean Bias | 27 | 50 | 94 |
| | RMSE | 421 | 326 | 304 |

*Competing interests.* We have no competing interests to declare.

*Acknowledgements.* PS, LS, and FM have received funding from the European Union's Horizon 2020 research and innovation programme under grant agreement No. 101003687 (PROVIDE). PS and FM have received funding from Austrian Climate Reserch Programm (ACRP) - 14th call, under grant agreement No. KR21KB0K00001 (HyMELT-CC). LS is recipient of a DOC Fellowship of the Austrian Academy of Sciences at the Department of Atmospheric and Cryospheric Sciences, Universität Innsbruck (No. 25928). This text reflects only the authors' view and the Agency is not responsible for any use that may be made of the information it contains.



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
