# Peer review of "Recent observations and glacier modeling point towards near complete glacier loss in western Austria (Ötztal and Stubai mountain range) if 1.5°C is not met"

_EGUsphere, 2024_

## Referee Comment (RC1)

Review of *Recent observations and glacier modeling point towards near complete glacier loss in western Austria (Ötztal and Stubai mountain range) if 1.5 °C is not met* by Hartl and Schmitt et al.

By Michael McCarthy

**General comments**

This study presents an improved observational dataset of recent glacier change in the Ötztal and Stubai mountain range and uses it with the large-scale glacier model OGGM to make projections of future glacier change in the region to the end of the century. OGGM is run in two different configurations: OGGM default and OGGM regional, where OGGM default uses globally available datasets for initialisation and calibration, and OGGM regional uses more accurate, local datasets in combination with an updated initialisation and calibration workflow. The OGGM projections are compared with each other and with projections from other large- and local-scale glacier models. The observational dataset consists of homogenised multi-temporal glacier outlines and volume estimates.

OGGM regional projects faster glacier decline in the Ötztal and Stubai mountain range than other large-scale glacier models, and suggests around 2.7% of glacier volume will remain by 2100 if climate warming is limited to 1.5 C. Under higher warming levels, it suggests glacier decline will occur faster, and remaining glacier volumes will be smaller by 2100. The observational dataset points to extensive glacier decline in the region in the last two decades.

The study is very clearly presented, appears to have been carefully implemented, and, in my view, makes at least three valuable contributions to the field:

1) By using more and higher quality data for initialisation and calibration and developing an updated initialisation and calibration workflow in OGGM regional, it addresses one of the key challenges of large-scale glacier models, which is that they tend to be over-parameterised due to a lack of observational data (e.g., Rounce et al., 2020). As such, it provides a framework for better regional-scale glacier modelling in the future.
2) By presenting new projections of glacier change in the Ötztal and Stubai mountain range, it adds to the evidence base around future glacier change in this region. It provides new and useful information about the impacts of different warming levels on regional-scale glacier evolution.
3) Via the improved observational dataset, it provides a more detailed understanding of recent glacier change in the region.

I have no major comments but think some minor modifications, listed below, could improve what is already a very good manuscript.

**Specific and technical comments**

Title and elsewhere in text: I don't suggest you change it, but is only 2.7% of 2017 glacier volume remaining by 2100 under 1.5 C not already 'near complete' glacier loss?

L45: Suggest remove 'between'.

L240 and throughout the text: Suggest 'evaluation' is preferable to 'validation', e.g., Oreskes et al (1994).

L303: What happens to the remaining 18 glaciers?

L311: I agree the model seems to reproduce the WGMS mass balance observations relatively well from 2000 onwards, over the calibration period. But it seems to produce more positive mass balances than the observations before 2000, which makes me wonder if it will produce too negative mass

balances in the future. I think this offset should be mentioned explicitly in the text, for transparency. Do you have any idea what might be causing it? Could it be the observations themselves, or are potentially more data still required for calibration? It could also be helpful to provide some performance metrics for the period before 2000 in a second table in the Appendix.

Section 3.3: My understanding is that all the results presented in this section are from OGGM regional. If so, it would be helpful to say this, e.g., on L314, 'The aggregated OGGM regional outcomes …'.

L327: The point about mitigation measures might be better in the Discussion than the Results?

L328: Does this increase in the number of individual glaciers account for fragmentation?

L338: On the topic of 'the full range of possible outcomes', it would be interesting to see the sensitivity of future volume change to the mean annual precipitation of the difference scenarios. Is this the major control on inter-scenario variability within warming levels?

L426: 'This is primarily due to their initialization strategy' seems like an assertion? 'This may be due to …' might be a better formulation.

Section 4.4.2: Earlier in the text, debris and avalanching are mentioned. These could be added explicitly here in relation to 'unresolved processes', as both are potential sources of error compensation and therefore calibration difficulties, and as both are starting to be considered in large-scale glacier models (e.g., Compagno et al. 2022).

Figure S4 caption: 'see Table ??' needs to be updated.

Figure S10 caption: Suggest 'For future projections except 1.5 C'.

Table S4 caption: Instead of 'supp. Figures S3 and S4' just 'Figures S3 and S4'.

**References**

Rounce, D. R., Khurana, T., Short, M. B., Hock, R., Shean, D. E., & Brinkerhoff, D. J. (2020). Quantifying parameter uncertainty in a large-scale glacier evolution model using Bayesian inference: application to High Mountain Asia. Journal of Glaciology, 66(256), 175-187.

Oreskes, N., Shrader-Frechette, K., & Belitz, K. (1994). Verification, validation, and confirmation of numerical models in the earth sciences. *Science*, *263*(5147), 641-646.

Compagno, L., Huss, M., Miles, E. S., McCarthy, M. J., Zekollari, H., Dehecq, A., ... & Farinotti, D. (2022). Modelling supraglacial debris-cover evolution from the single-glacier to the regional scale: an application to High Mountain Asia. The Cryosphere, 16(5), 1697-1718.

---

## Referee Comment (RC2)

**Review of "Recent observations and glacier modeling point towards near complete glacier loss in western Austria (Ötztal and Stubai mountain range) if 1.5°C is not met"**

The study by Hartl, Schmitt, and colleagues uses a new version of the OGGM glacier model, called OGGM-regional, to make updated glacier volume predictions for the Ötztal and Stubai regions. They calibrated and validated this model using new glacier area and volume data and a new calibration procedure described in Aguayo et al. (2023). Furthermore, they compare their updated simulations with the original version of OGGM (default) as well as with results from other regional/global glacier model studies (e.g., Zekollari et al., 2019; Cook et al., 2023).

The study finds that glaciers in the Ötztal and Stubai regions may decline faster than projected by other large-scale glacier models. It estimates that if global warming is limited to +1.5°C, about 2.7% of glacier volume will remain by 2100 (so actually almost everything disappears). With higher warming levels (2-4°C), glaciers could disappear by or before 2100. The model's results align closely with observed mass balance data from WGMS (especially after 2000) and area changes from 1997 to 2006 (not used in calibration), indicating strong performance of OGGM-regional and showing reliability + a clear step forward in regional glacier modelling.

In my view, the study is well-presented, well-written, detailed, and scientifically intriguing. It uses high-quality data for initialization and calibration, highlighting the challenges in calibrating large-scale glacier models, which often lack enough observational data (and model structure). Including more data, as done here, improves clearly the performance of these large-scale models.

While the results provide new, more pessimistic estimates of how regional glaciers in western Austria will respond to warming, the study's approach and methodology are even more compelling to me. I also really like the approach of modelling the glaciers using different warming levels, rather than under specific SSP scenarios.

I only have some **textual comments/suggestions** which I hope could help the authors to finalise their paper:

- The title is very compelling. Well done!
- Line 6: Suggestion: different climate scenarios -> different warming levels?
- Line 9-11: I did not completely get this extrapolation. In the abstract, it seems that the area/volume change rate is extrapolated, but further on in the paper, I think it is more the average observed surface elevation changes that are extrapolated, right? I find that this gets also a little bit too much attention in the abstract (which I find in general very well written)
- Line 20: Under +2°C, 0.4% remains, so I would expect that under +2.7°C, somehow 0.1 or 0.2% would remain? "Less than 1%" seems a bit too positive
- Line 27: Example of these local factors? Do you mean debris?
- Line 46-47: Reference?
- Line 31: What do you mean with m of elevation per year? I would stay with the standard unit of m w.e. per year?
- Line 33: ... losses observed for smaller glaciers

- Line 45: cannot -> could not
- Line 64: Any reason why you not directly mention here OGGM and then also refer to Maussion et al. (2019)?
- Line 67: driven by?
- Line 70: Border with? Border on sounds a bit strange
- Figure 1:
  - The grey lines in the lower inset are hardly visible. Consider using a different (stronger, more contrasting) colour.
  - Add m a.s.l. to the scale bar for the elevation.
  - Can you increase the upper inset a bit to show entire Austria?
  - Add a north arrow to the plot
  - I also suggest to use panel labels (e.g., a-b-c)
- Line 84: This sentence needs a reference to a study quantifying this evolution
- Line 85: The abbreviation DEM has already been declared in the introduction
- Line 100: replace ";" by a ","
- Line 103: larger region studies -> regional/global studies?
- Line 104: Not extremely important, but suggest to use the same order of the mentioned studies as in line 100
- Line 116: DEM abbreviation has been used before (2x)
- Line 118: Is this density the density in the study regio or in general?
- Table 3: Why is Daniel Farinotti mentioned twice?
- Line 139: Maybe add this information in line 128? … using ArcMap GIS software
- Line 143-146: I do not completely get this part. Do you mean exclusion instead of inclusion?
- Line 147-149: Repetition of what was sad before?
- Line 152: Did you yourself apply this multi-person mapping approach?
- Line 164-165: This sentence could be removed to save some space
- Line 190: How was the ice volume determined for glaciers without measurements?
- Line 202: No need to refer here to Zekollari et al. (2024) in my opinion. OGGM is described in Maussion et al. (2019;2023), right?
- Line 214: What do you mean with no further volume or area thresholds?
- Line 247: regional data = regional ice thickness observations?
- Line 256-259 -> super interesting! So having 2 area inventories allows to calibrate this parameter
- Line 269: Since 2006 -> between 2006 and 2017/18
- Figure 3:
  - Panel b and c-> Replace m/a to m yr$^{-1}$ (which you use in the text and caption)
  - The labels are all pretty small, you might try out to make the figure/labels a bit larger
- Line 276: Interesting -> maybe refer to Kneib et al. 2024?
- Line 303: Why are the other glaciers not working?
- Figure 4: Again pretty small labels, try to increase the size
- Figure 5: Increase the size of this figure, for me this is one of the key results of the paper -> area is way better matched by OGGM-regional
- Line 318: until beyond -> by?
- Line 337: Don't -> do not

- Figure 6 -> make Figure much larger (very little detail can be see now with the figure being so small)
- Line 349-350: This is a bit contra-intuitive for me. As glacier mass is lost, the small glaciers are lost faster, so their contribution decreases?
- Line 363: Maybe note here that Kneib et al. (2024) show the contribution and impact of such avalanching on Argentiere glacier.
- Line 382: What do you mean with observed area 1979?
- Line 383: "compared to … " could be removed, as this is rather logic
- Line 390-391: very interesting finding!
- Line 420: this is rather logic when comparing +1 to +2°C with warmer scenarios… I guess the comparison with Zekollari 2019 is a more valid comparison for GloGEMflow
- Line 426: Is the main result not related to the fact that Cook et al. (2023) uses Hugonnet's mass balance as is for future projections and thus shows committed mass loss more than future projections?
- Line 433-435: For which scenario is this?
- Line 439: Do you think taking into account more complex dynamics can results in more volume remaining? i.e., that flowline models show too fast ice losses (although differences are small)
- Line 487: Could remove region of interest (ROI) in this sentence

---

## Author Comment (AC1)

We sincerely thank Michael McCarthy and Lander Van Tricht for their encouraging feedback and the constructive comments! We very much appreciate the time the reviewers and editor have spent on the manuscript and will be happy to address the comments. Please find responses to specific points below in blue text.

**Review by Michael McCarthy**

General comments

This study presents an improved observational dataset of recent glacier change in the Ötztal andStubai mountain range and uses it with the large-scale glacier model OGGM to make projections of future glacier change in the region to the end of the century. OGGM is run in two different configurations: OGGM default and OGGM regional, where OGGM default uses globally available datasets for initialisation and calibration, and OGGM regional uses more accurate, local datasets in combination with an updated initialization and calibration workflow. The OGGM projections are compared with each other and with projections from other large- and local-scale glacier models. The observational dataset consists of homogenised multi-temporal glacier outlines and volume estimates.

OGGM regional projects faster glacier decline in the Ötztal and Stubai mountain range than other large-scale glacier models, and suggests around 2.7% of glacier volume will remain by 2100 if climate warming is limited to 1.5 C. Under higher warming levels, it suggests glacier decline will occur faster, and remaining glacier volumes will be smaller by 2100. The observational dataset points to extensive glacier decline in the region in the last two decades.

The study is very clearly presented, appears to have been carefully implemented, and, in my view, makes at least three valuable contributions to the field:
1) By using more and higher quality data for initialisation and calibration and developing an updated initialisation and calibration workflow in OGGM regional, it addresses one of the key challenges of large-scale glacier models, which is that they tend to be over-parameterised due to a lack of observational data (e.g., Rounce et al., 2020). As such, it provides a framework for better regional-scale glacier modelling in the future.
2) By presenting new projections of glacier change in the Ötztal and Stubai mountain range, it adds to the evidence base around future glacier change in this region. It provides new and useful information about the impacts of different warming levels on regional-scale glacier evolution.
3) Via the improved observational dataset, it provides a more detailed understanding of recent glacier change in the region.

I have no major comments but think some minor modifications, listed below, could improve what is already a very good manuscript.
Thank you!

Specific and technical comments

Title and elsewhere in text: I don't suggest you change it, but is only 2.7% of 2017 glacier volume remaining by 2100 under 1.5 C not already 'near complete' glacier loss?
Yes, 2.7% of the 2017 volume is "near complete" glacier loss in practical terms. We discussed how to present these numbers and settled on the current phrasing since we feel it best reflects the model output. While the glaciers will largely be gone by the end of the century even in the 1.5°C scenario, we feel that the "near complete glacier loss" phrasing allows for the possibility of small ice bodies in sheltered or otherwise topographically favoured locations to remain beyond the end of the century.

L45: Suggest remove 'between'.
Changed as suggested
L240 and throughout the text: Suggest 'evaluation' is preferable to 'validation', e.g., Oreskes et al (1994).
Changed throughout the text as suggested

L303: What happens to the remaining 18 glaciers?
The 18 glaciers for which OGGM Regional was unable to perform a successful model run were omitted from further analysis. The primary issue with these glaciers was that they were too small to create an elevation band flowline. We added the following sentence to the text for clarification: "The 18 glaciers that could not be successfully simulated, mainly due to their limited elevation range, are excluded from further analysis."

L311: I agree the model seems to reproduce the WGMS mass balance observations relatively well from 2000 onwards, over the calibration period. But it seems to produce more positive mass balances than the observations before 2000, which makes me wonder if it will produce too negative mass balances in the future. I think this offset should be mentioned explicitly in the text, for transparency.  Do you have any idea what might be causing it? Could it be the observations themselves, or are potentially more data still required for calibration? It could also be helpful to provide some performance metrics for the period before 2000 in a second table in the Appendix.
Thank you for pointing this out. Before the year 2000, the model had no information about the past glacier state, resulting in many possible trajectories to reach the observed glacier state (e.g. Eis et al., 2019). This can be partly attributed to the diffusive nature of glacier dynamics. Therefore, the further we go back in time from the model's first 'observed' state (2000 for OGGM default and 2006 for OGGM Regional), the more we deviate from the actual past and instead follow one of many possible trajectories. Additional observations could help constrain the model trajectory to align more closely with actual historical events.

To be transparent about this limitation, we added the following sentences: 'However, the further back in time we look from 2000, the larger the discrepancies between the models and observations become. This is due to the diffusive nature of glacier dynamics, and in the absence of past observations, the model selects one possible trajectory among many possibilities (e.g. Eis et al., 2019).'

Section 3.3: My understanding is that all the results presented in this section are from OGGM

regional. If so, it would be helpful to say this, e.g., on L314, 'The aggregated OGGM regional outcomes …'.

Changed as suggested

L327: The point about mitigation measures might be better in the Discussion than the Results?
Yes, thank you. Moved to discussion and rephrased slightly.

L328: Does this increase in the number of individual glaciers account for fragmentation?
No, fragmentation is not incorporated in the analysis. We have added this information to the text to clarify.

L338: On the topic of 'the full range of possible outcomes', it would be interesting to see the sensitivity of future volume change to the mean annual precipitation of the difference scenarios. Is this the major control on inter-scenario variability within warming levels?
Even when using temperature levels for aggregation, we observe a significant spread in individual temperatures within the defined temperature levels. Figure 1 shows the mean annual temperature for melt (T_melt) and the mean annual solid precipitation (Prcp_solid) across all model runs plotted against total glacier volume. T_melt and Prcp_solid are defined such that the annual Mass Balance (MB) can be calculated as:

**MB = Prcp_solid - T_melt * melt_factor.**

The figure highlights a spread in both precipitation and temperature, demonstrating that the range of possible outcomes is driven by a combination of both variables. For instance, model runs that project a larger glacier volume in 2100 tend to have both a smaller T_melt and a larger Prcp_solid (as solid precipitation is also a function of temperature), and vice versa.

[Figure]

*Figure 1: Left column shows annual mean temperature for melt and the right column annual mean solid precipitation on the x-axis, relative to the total volume on the y-axis, for individual model projections of OGGM regional. The individual models are colored, depending on their aggregated temperature level.*

L426: 'This is primarily due to their initialization strategy' seems like an assertion? 'This may be due to …' might be a better formulation.

This is really due to their initialization strategy, as explained in the second part of the sentence. Their initialization strategy differs significantly from the other models, resulting in a larger volume in 2020. To improve clarity, we removed the term 'primarily' from the original text. See also answer to Reviewer#2.

Section 4.4.2: Earlier in the text, debris and avalanching are mentioned. These could be added explicitly here in relation to 'unresolved processes', as both are potential sources of error compensation and therefore calibration difficulties, and as both are starting to be considered in largescale glacier models (e.g., Compagno et al. 2022).

Yes, good point. We have added the following text:

"OGGM also does not explicitly account for the impacts of debris cover or avalanches on mass balance. These processes are increasingly being incorporated in glacier modeling at local and larger scales with promising results for improved representation of mass balance distribution (e.g., Compagno et al, 2022, Kneib et al, 2024)"

Figure S4 caption: 'see Table ??' needs to be updated.

Fixed.

Figure S10 caption: Suggest 'For future projections except 1.5 C'.

Changed as suggested, thanks.

Table S4 caption: Instead of 'supp. Figures S3 and S4' just 'Figures S3 and S4'.

Adjusted.

References

Rounce, D. R., Khurana, T., Short, M. B., Hock, R., Shean, D. E., & Brinkerhoff, D. J. (2020). Quantifying parameter uncertainty in a large-scale glacier evolution model using Bayesian inference: application to High Mountain Asia. Journal of Glaciology, 66(256), 175-187.

Oreskes, N., Shrader-Frechette, K., & Belitz, K. (1994). Verification, validation, and confirmation of numerical models in the earth sciences. Science, 263(5147), 641-646.

Compagno, L., Huss, M., Miles, E. S., McCarthy, M. J., Zekollari, H., Dehecq, A., ... & Farinoti, D.(2022). Modelling supraglacial debris-cover evolution from the single-glacier to the regional scale: an application to High Mountain Asia. The Cryosphere, 16(5), 1697-1718.

---

## Author Comment (AC2)

We sincerely thank reviewers Michael McCarthy and Lander Van Tricht for their encouraging feedback and the constructive comments! We very much appreciate the time the reviewers and editor have spent on the manuscript and will be happy to address the comments. Please find responses to specific points below in blue text.

**Review by Lander Van Tricht**

The study by Hartl, Schmitt, and colleagues uses a new version of the OGGM glacier model, called OGGM-regional, to make updated glacier volume predictions for the Ötztal and Stubai regions. They calibrated and validated this model using new glacier area and volume data and a new calibration procedure described in Aguayo et al. (2023). Furthermore, they compare their updated simulations with the original version of OGGM (default) as well as with results from other regional/global glacier model studies (e.g., Zekollari et al., 2019; Cook et al., 2023).

The study finds that glaciers in the Ötztal and Stubai regions may decline faster than projected by other large-scale glacier models. It estimates that if global warming is limited to +1.5°C, about 2.7% of glacier volume will remain by 2100 (so actually almost everything disappears). With higher warming levels (2-4°C), glaciers could disappear by or before 2100. The model's results align closely with observed mass balance data from WGMS (especially a]er 2000) and area changes from 1997 to 2006 (not used in calibration), indicating strong performance of OGGM-regional and showing reliability + a clear step forward in regional glacier modelling.

In my view, the study is well-presented, well-wri4en, detailed, and scientifically intriguing. It uses high-quality data for initialization and calibration, highlighting the challenges in calibrating large-scale glacier models, which o]en lack enough observational data (and model structure). Including more data, as done here, improves clearly the performance of these large-scale models.

While the results provide new, more pessimistic esAmates of how regional glaciers in western Austria will respond to warming, the study's approach and methodology are even more compelling to me. I also really like the approach of modelling the glaciers using different warming levels, rather than under specific SSP scenarios.

I only have some textual comments/suggestions which I hope could help the authors to finalise their paper:

- The Title is very compelling. Well done!

Thank you!

- Line 6: Suggestion: different climate scenarios -> different warming levels?

Changed as suggested

- Line 9-11: I did not completely get this extrapolation. In the abstract, it seems that the area/volume change rate is extrapolated, but further on in the paper, I think it is more the average observed surface elevation changes that are extrapolated, right? I find that this gets also a little bit too much attention in the abstract (which I find in general very well written)

We rephrased this as follows aiming to reduce the focus on the extrapolation:

"The model projections for all scenarios predict a faster glacier decline than a constant change scenario based on the observed change rates for 2006 to 2017. This highlights the need for dynamic, climate-aware glacier models to quantify the range of possible futures and trajectories to deglaciation."

We used the change extracted from DEM differencing and divided it by the number of years between DEM acquisitions to get a yearly change rate. For each pixel in the ROI, we linearly extrapolated that change rate into the future as long as there is ice in the pixel (ice thickness grid from Helfricht et al,, 2019).

- Line 20: Under +2°C, 0.4% remains, so I would expect that under +2.7°C, somehow 0.1 or 0.2% would remain? "Less than 1%" seems a bit too positive

The first statement (0.4% under 2°C warming) refers to a particular glacier, which we highlight because it is the largest glacier in our study region. "Less than 1%" refers to the entire study region. We have slightly restructured the order of the sentences to clarify this.

- Line 27: Example of these local factors? Do you mean debris?

We intend this as a general statement that encompasses all possible factors. Debris is one of numerous factors that can play a role at the scale of individual glaciers. Local mass balance variability tends to be related to local glacier geometry and/or topography, wind patterns, avalanches, etc. We feel that Brun et al (2017) showcase this well and cited that study here for this reason. We would prefer to keep this statement as is for conciseness in the introduction but can of course add some examples of local factors if the reviewer or editor feel strongly about it.

- Line 46-47: Reference?

Thank you for spotting, we added Marzeion et al. 2020 as reference.

- Line 31: What do you mean with m of elevation per year? I would stay with the standard unit of m w.e. per year?

Fixed.

- Line 33: … losses observed for smaller glaciers

Changed as suggested

- Line 45: cannot -> could not

Changed as suggested

- Line 64: Any reason why you not directly mention here OGGM and then also refer to Maussion et al. (2019)?

Added OGGM and citation.

- Line 67: driven by?

Changed to "driven by"

- Line 70: Border with? Border on sounds a bit strange

We think "border on" is correct usage here
(https://www.merriam-webster.com/dictionary/border%20on)

- Figure 1:
• The grey lines in the lower inset are hardly visible. Consider using a different (stronger, more contrasAng) colour.
• Add m a.s.l. to the scale bar for the elevation.
• Can you increase the upper inset a bit to show entire Austria?
• Add a north arrow to the plot

• I also suggest to use panel labels (e.g., a-b-c)

Adjusted the figure as suggested: Changed color of the lines to orange instead of grey, added m a.s.l. to the scale bar, added north arrow, added panel labels, adjusted the overview inset to show all of Austria.

- Line 84: This sentence needs a reference to a study quantifying this evolution

Added reference (Fischer et al 2015) and point readers to Table 1 for additional sources.

Fischer, A., Seiser, B., Stocker Waldhuber, M., Mitterer, C., & Abermann, J. (2015). Tracing glacier changes in Austria from the Little Ice Age to the present using a lidar-based high-resolution glacier inventory in Austria. *The Cryosphere*, *9*(2), 753-766.

- Line 85: The abbreviation DEM has already been declared in the introduction

Adjusted.

- Line 100: replace ";" by a ","

Fixed

- Line 103: larger region studies -> regional/global studies?

Changed sentence to: "Compared to the regional estimate of ice volume in the ROI by Helfricht et al 2019, the estimates in the global and regional studies of Farinotti et al 2019, Millan et al 2022, and Cook et al 2023…."

- Line 104: Not extremely important, but suggest to use the same order of the mentioned studies as in line 100

Changed as suggested

- Line 116: DEM abbreviation has been used before (2x)

Removed here and above.

- Line 118: Is this density the density in the study region or in general?

This is the point density as stated for high mountain areas in the report produced by the government agency for geodata that carried out the survey and processed the point clouds. It is specific to their area classifications but the point density generally does not vary very much. It goes up to a bit over 7 on average for settlement areas.

- Table 3: Why is Daniel Farinottis mentioned twice?

Thank you for spotting this. The citation was mentioned twice because we cited both the publication and the data separately. For consistency, we have removed the citation to the data for Farinotti and Helfricht and retained only the citation to the publications, as was already done for Millan and Cook.

- Line 139: Maybe add this information in line 128? … using ArcMap GIS software

Added in L 128, removed in L 139.

- Line 143-146: I do not completely get this part. Do you mean exclusion instead of Inclusion?

We used a DEM of differences (surface elevation change) as part of the mapping process based on the assumption that the data will show surface change if there is still ice. For ice bodies that melted between the acquisitions in 2006 and 2017, surface change can still be apparent in the data even though the feature was gone by 2017. We are saying that there may be a small number of cases where ice features that melted between 2006 and 2017 are still included in the inventory due to this effect.

- Line 147-149: Repetition of what was said before?)

No - previously we explain that features that still show surface elevation change were included (see comment above). Here, we note that former glaciers that no longer show elevation change were excluded.

- Line 152: Did you yourself apply this multi-person mapping approach?

We did not, but we follow the approach of Abermann et al exactly and in the same region and expect the estimated uncertainty range to be applicable in this case in the same way as in the prior works of Abermann et al. We have added a reference to Abermann et al 2010 here for improved clarity.

- Line 164-165: This sentence could be removed to save some space

Agreed and removed.

- Line 190: How was the ice volume determined for glaciers without measurements?

The ice volume was computed using the model of Huss & Farinotti (2012), which was calibrated with the measured data. For the glaciers without measurements the result is the output of the calibrated model. We added a citation of Huss & Farinotti (2012) here for clarity.

- Line 202: No need to refer here to Zekollari et al. (2024) in my opinion. OGGM is described in Maussion et al. (2019;2023), right?

We cited Zekollari 2024 here, as it is a peer-reviewed paper that provides a more detailed description of OGGM v1.6.1. In contrast, Maussion 2023 refers only to the code and online documentation.

- Line 214: What do you mean with no further volume or area thresholds?

Rephrased for clarity and removed the mention of thresholds. We originally experimented with different ways of setting size thresholds. This phrase was left over but it is not needed.

- Line 247: regional data = regional ice thickness observations?

Thanks for pointing out this inaccuracy. The regional data consists of a volume estimate and a volume change observation. For clarity, we have adapted the sentence to: "OGGM default utilizes global datasets for calibration and initialization (using one glacier outline), while OGGM regional uses regional data, including one volume estimation, one volume change observation, and two glacier outlines at different timestamps."

- Line 256-259 -> super interesting! So having 2 area inventories allows to calibrate this parameter

Yes, indeed, two outlines could help with this. Currently, we are doing this on a regional basis, but for future applications, it could be considered on a glacier-specific basis.

- Line 269: Since 2006 -> between 2006 and 2017/18

Changed as suggested

- Figure 3:

• Panel b and c-> Replace m/a to m yr-1 (which you use in the text and caption)

• The labels are all pretty small, you might try out to make the figure/labels a bit Larger

Adapted units and label size.

- Line 276: Interesting -> maybe refer to Kneib et al. 2024?

Added citation as suggested.

- Line 303: Why are the other glaciers not working?

Same answer as for Reviewer#1: The 18 glaciers for which OGGM Regional was unable to perform a successful model run were omitted from further analysis. The primary issue with these

glaciers was that they were too small to create an elevation band flowline. We added the following sentence to the text for clarification: "The 18 glaciers that could not be successfully simulated, mainly due to their limited elevation range, are excluded from further analysis."

- Figure 4: Again pretty small labels, try to increase the size

Adjusted labels size.

- Figure 5: Increase the size of this figure, for me this is one of the key results of the paper -> area is way better matched by OGGM-regional

We increased the height and font size of the figure while keeping the width unchanged to be consistent with the journal's latex template.

- Line 318: until beyond -> by?

Changed to "beyond" (removed until)

- Line 337: Don't -> do not

fixed

- Figure 6 -> make Figure much larger (very little detail can be see now with the figure being so small)

We increased the height and font size of the figure while keeping the width unchanged to be consistent with the journal's latex template.

- Line 349-350: This is a bit contra-intuitive for me. As glacier mass is lost, the small glaciers are lost faster, so their contribution decreases?

We are discussing the **relative** contribution of small glaciers to total glacier area and volume in the region. The idea here is large glaciers are turning into small glaciers as they lose mass. Regionally, the relative contribution of the smaller ones compared to the large ones increases, simply because there are no or only very few large glaciers left. This is relevant in our study area because we have mostly small and very small glaciers. The point we are trying to make is that it is important to also account for the small glaciers in modeling and obs.

- Line 363: Maybe note here that Kneib et al. (2024) show the contribution and impact of such avalanching on Argentiere glacier.

Added citations of Kneib et al (2024 a, b) and Hynek et al (2024)

Kneib, M., Dehecq, A., Gilbert, A., Basset, A., Miles, E. S., Jouvet, G., ... & Six, D. (2024). Distributed surface mass balance of an avalanche-fed glacier. *EGUsphere*, *2024*, 1-35.

Kneib, M., Dehecq, A., Brun, F., Karbou, F., Charrier, L., Leinss, S., ... & Maussion, F. (2024). Mapping and characterization of avalanches on mountain glaciers with Sentinel-1 satellite imagery. *The Cryosphere*, *18*(6), 2809-2830.

Hynek, B., Binder, D., Citterio, M., Hillerup Larsen, S., Abermann, J., Verhoeven, G., ... & Schöner, W. (2024). Accumulation by avalanches as a significant contributor to the mass balance of a peripheral glacier of Greenland. *The Cryosphere*, *18*(11), 5481-5494.

- Line 382: What do you mean with observed area 1979?

Thank you for spotting this, this was an accidental typo and the numbers were swapped. It should be the 'observed area 1997'. We changed this mistake in the manuscript.

- Line 383: "compared to … " could be removed, as this is rather logic

Agreed and removed.

- Line 390-391: very interesting finding!

Thank you, we think so too!

- Line 420: this is rather logic when comparing +1 to +2°C with warmer scenarios… I guess the comparison with Zekollari 2019 is a more valid comparison for GloGEMflow

Agree, the comparison to Zekollari 2019 is done in L416-L417. We shortened the discussion about Compagno 2021 in the manuscript and merged the two sentences:
"Compagno (2021) limited future warming to +1°C to +2°C degrees in their projections. Their glacier evolution trajectory is comparable to OGGM default and Rounce et al (2023) until about mid-century and diverges to a more glacier-favorable outlook thereafter, which is to be expected when only running cooler scenarios as in Compagno (2021)  (+1°C to +2°C) and comparing it to warmer scenarios for OGGM and Rounce et al (2023) (+1.5°C to +4.0°C)."
to
"Compagno (2021)'s study shows similar glacier evolution to OGGM default and Rounce et al (2023) until mid-century but diverges to a more glacier-favorable trend thereafter, as expected when comparing cooler scenarios (+1°C to +2°C) with warmer ones (+1.5°C to +4.0°C)."

- Line 426: Is the main result not related to the fact that Cook et al. (2023) uses Hugonnet's mass balance as is for future projections and thus shows committed mass loss more than future projections?

Yes, you are correct that Cook et al. (2023) demonstrates a committed mass loss rather than future projections. However, the difference in glacier volume by 2050 is primarily related to their initialization strategy, which results in starting from a larger volume in 2020. Notably, the volume change rates between 2025 and 2050 are larger for Cook et al. (2023) (-0.9 km³ yr⁻¹) compared to OGGM Regional (-0.7 km³ yr⁻¹). To clarify this point, we added the following to the manuscript: "As a result, the projected glacier volume for 2050 in Cook et al. (2023) remains larger than that of OGGM regional, despite their model predicting a higher absolute volume change rate between 2025 and 2050 (-0.9 km³ yr⁻¹ for Cook et al., 2023, vs. -0.7 km³ yr⁻¹ for OGGM regional)."

- Line 433-435: For which scenario is this?

It is the median for all scenarios used by Hanzer 2018. For clarity we added the following:
"Comparing the median volume **of all scenarios** of Hanzer et al. (2018)'s simulations …"

- Line 439: Do you think taking into account more complex dynamics can results in more volume remaining? i.e., that flowline models show too fast ice losses (although differences are small)

Determining the influence of dynamics on overall evolution is challenging without conducting experiments under controlled conditions. For instance, driving both a distributed model and a flowline model with the same mass balance would provide the necessary basis for comparison. Without such controlled experiments, it remains difficult to assess the specific impact of dynamics on the system's overall behavior. We are not aware of such a study, especially at these small glacier sizes.

However, our intuition here (and observations from the field) points towards glacier dynamics mattering less as glaciers retreat, and that the factors determining whether ice will remain in some areas will be related to local factors more than dynamics. Topographic shadows, avalanches, and debris cover come to mind.

- Line 487: Could remove region of interest (ROI) in this sentence
Removed.

References:

- Abermann, J., Fischer, A., Lambrecht, A., and Geist, T.: On the potential of very high-resolution repeat DEMs in glacial and periglacial environments, The Cryosphere, 4, 53–65, https://doi.org/10.5194/tc-4-53-2010, 2010.
- Eis, J., Maussion, F., and Marzeion, B.: Initialization of a global glacier model based on present-day glacier geometry and past climate information: an ensemble approach, The Cryosphere, 13, 3317–3335, doi:10.5194/tc-13-3317-2019, 2019.
- Huss, M., and Farinotti, D. (2012). Distributed ice thickness and volume of all glaciers around the globe. J. Geophys. Res. 117:F04010. doi: 10.1029/2012JF002523
- Marzeion, B., Hock, R., Anderson, B., Bliss, A., Champollion, N., Fujita, K., Huss, M., Immerzeel, W., Kraaijenbrink, P., Malles, J., Maussion, F., Radić, V., Rounce, D. R., Sakai, A., Shannon, S., Wal, R. and Zekollari, H.: Partitioning the Uncertainty of Ensemble Projections of Global Glacier Mass Change, Earth's Futur., 8(7), doi:10.1029/2019ef001470, 2020.